# Online Unsupervised Learning of Visual Representations and Categories

## Abstract

Real world learning scenarios involve a nonstationary distribution of classes with sequential dependencies among the samples, in contrast to the standard machine learning formulation of drawing samples independently from a fixed, typically uniform distribution. Furthermore, real world interactions demand learning on-the-fly from few or no class labels. In this work, we propose an unsupervised model that simultaneously performs online visual representation learning and few-shot learning of new categories without relying on any class labels. Our model is a prototype-based memory network with a control component that determines when to form a new class prototype. We formulate it as an online mixture model, where components are created with only a single new example, and assignments do not have to be balanced, which permits an approximation to natural imbalanced distributions from uncurated raw data. Learning includes a contrastive loss that encourages different views of the same image to be assigned to the same prototype. The result is a mechanism that forms categorical representations of objects in nonstationary environments. Experiments show that our method can learn from an online stream of visual input data and its learned representations are significantly better at category recognition compared to state-of-the-art self-supervised learning methods.

## 1 Introduction

Humans operating in the real world have the opportunity to learn from large quantities of unlabeled data. However, as an individual moves within and between environments, the stream of experience has complex temporal dependencies. The goal of our research is to tackle the challenging problem of online unsupervised representation learning in the setting of environments with naturalistic structure. We wish to design learning algorithms that facilitate the categorization of objects as they are encountered and re-encountered. In representation learning, methods are often evaluated based on their ability to classify from the representation using either supervised linear readout or unsupervised clustering over the full dataset, both of which are typically done in a separate post-hoc evaluation phase. Instead, a key aim of our work is to predict object categories throughout training and evaluation, where categorization is performed by grouping a new instance with one or more previous instances, and does not rely on externally provided labels at any stage.

Unsurprisingly, the structure of natural environments contrasts dramatically with the standard scenario typically assumed by many machine learning algorithms: mini-batches of independent and identically distributed (iid) samples from a well-curated dataset. In unsupervised visual representation learning, the most successful methods rely on iid samples. Contrastive-based objectives (Chen et al., 2020a; He et al., 2020) typically assume that each instance in the mini-batch forms its own instance class. When this assumption is violated due to autocorrelations in a naturalistic online streaming setting, contrastive approaches will push same-class instances apart. Clustering-based learning frameworks (Caron et al., 2018; Asano et al., 2020; Caron et al., 2020) have their own difficulties in environments with nonstationary and imbalanced class distributions: they assume that the set of cluster centroids remain relatively stable and that the clusters are balanced in size.

To make progress on the challenge of unsupervised visual representation learning and categorization in a naturalistic setting, we propose the *online unsupervised prototypical network (OUPN)*, which performs

learning of visual representations and object categories simultaneously in a single-stage process. Class prototypes are created via an online clustering procedure, and a contrastive loss (Chopra et al., 2005; van den Oord et al., 2018) is used to encourage different views of the same image to be assigned to the same cluster. Notably, our online clustering procedure is more flexible relative to other clustering-based representation learning algorithms, such as DeepCluster (Caron et al., 2018) and SwAV (Caron et al., 2020): OUPN performs learning and inference as an online Gaussian mixture model, where clusters can be created online with only a single new example, and cluster assignments do not have to be balanced, which permits an approximation to natural imbalanced distributions from uncurated raw data.

We train and evaluate our algorithm on a recently proposed naturalistic dataset, RoamingRooms (Ren et al., 2021), which uses imagery collected from a virtual agent walking through different rooms, and SAYCam (Sullivan et al., 2022), which is collected from head-mounted camera recordings from human babies. We compare to a suite of state-of-the-art self-supervised representation learning methods: SimCLR (Chen et al., 2020a), SwAV (Caron et al., 2020), and SimSiam (Chen & He, 2021). OUPN performs relatively well, as these methods are designed for batches of iid data and degrade significantly with non-iid streams. But even when we train these methods in an offline fashion—by shuffling the data to be iid—they underperform OUPN, which handles better the underlying data imbalance and exploits structure in the online temporal streams. In addition, we use RoamingOmniglot (Ren et al., 2021) as a benchmark, and also investigate the effect of imbalanced classes; we find that OUPN is very robust to an imbalanced distribution of classes. For a version of ImageNet with non-iid structure, RoamingImageNet, OUPN again outperforms self-supervised learning baselines when using matched batch sizes. These experiments indicate that OUPN supports the emergence of visual understanding and category formation of an online agent operating in an embodied environment.

## 2 Related Work

**Self-supervised learning.** Self-supervised learning methods discover rich and informative visual representations without class labels. *Instance-based approaches* aim to learn invariant representations of each image under different transformations (van den Oord et al., 2018; Misra & van der Maaten, 2020; Tian et al., 2020; He et al., 2020; Chen et al., 2020a;b; Grill et al., 2020; Chen & He, 2021; Assran et al., 2021). They typically work well with iid data and large batch sizes, which contrasts with realistic learning scenarios. Our method is also related to *clustering-based approaches*, which obtain clusters on top of the learned embedding and use the cluster assignments to constrain the embedding network. To compute the cluster assignment, DeepCluster (Caron et al., 2018; Zhan et al., 2020) and PCL (Li et al., 2021) use the $k$-means algorithm whereas SeLa (Asano et al., 2020) and SwAV (Caron et al., 2020) uses the Sinkhorn-Knopp algorithm (Cuturi, 2013). However, they typically assume a fixed number of clusters, and Sinkhorn-Knopp further assumes a balanced assignment as an explicit constraint. In contrast, our online clustering procedure is more flexible: it can create new clusters on-the-fly with only a single new example and does not assume balanced cluster assignments. Self-supervised pretraining or joint training has proven beneficial for online continual learning tasks (Zhang et al., 2020; Gallardo et al., 2021; Cha et al., 2021). There are works that apply self-supervised learning on datasets that have imbalanced classes with a focus to perform semantic and instance segmentation as downstream tasks (Cho et al., 2021; Van Gansbeke et al., 2021; Xiong et al., 2021; Hénaff et al., 2021; Hamilton et al., 2022; Vobecky et al., 2022). However, learning is still offline in these prior works. By contrast, here we aim to perform learning from an online stream of unlabeled images.

**Representation learning from video.** There has also been a surge of interest in leveraging video data to learn visual representations (Wang & Gupta, 2015; Pathak et al., 2017; Orhan et al., 2020; Zhu et al., 2020; Xiong et al., 2021). These approaches all sample video subsequences uniformly over the entire dataset, whereas our model directly learns from an online stream of data. Our model also does not have the assumption that inputs must be adjacent frames in the video.

**Online and incremental representation learning.** Our work is also related to online and continual representation learning (Rebuffi et al., 2017; Castro et al., 2018; Rao et al., 2019; Jerfel et al., 2019; Javed & White, 2019; Hayes et al., 2020). Continual mixture models (Rao et al., 2019; Jerfel et al., 2019) designate a categorical latent variable that can be dynamically allocated for a new environment. Our model has a

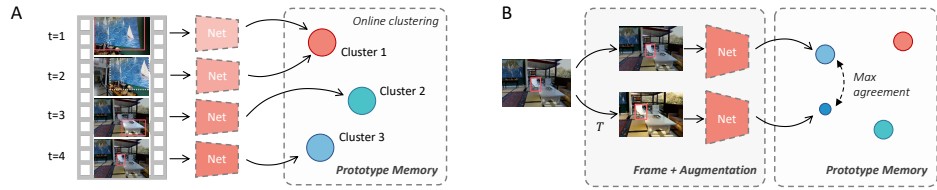

Figure 1: Our proposed online unsupervised prototypical network (OUPN). **A:** OUPN learns directly from an online visual stream. Images are processed by a deep neural network to extract representations. Representations are stored and clustered in a prototype memory. Similar features are aggregated in a cluster and new clusters can be dynamically created if the current feature vector is different from all existing clusters. **B:** The network learning uses self-supervision that encourages different augmentations of the same frame to have consistent cluster assignments.

similar mixture latent variable setup but one major difference is that we operate on example-level rather than task-level. Streaming learning (Hayes et al., 2019; 2020) aims to perform representation learning online. Most work here except Rao et al. (2019) assumes a fully supervised setting. Our prototype memory also resembles a replay buffer (Buzzega et al., 2020; Kim et al., 2020), but we store the feature prototypes instead of the inputs.

**Latent variable modeling on sequential data.** Our model also relates to a family of latent variable generative models for sequential data (Krishnan et al., 2015; Johnson et al., 2016; He et al., 2018; Denton & Fergus, 2018; Zhu et al., 2020). Like our model, these approaches aim to infer latent variables with temporal structure, but they use an input reconstruction criterion.

**Online mixture models.** Our clustering module is related to the literature on online mixture models, e.g., Carpenter & Grossberg (1987); Anderson (1991); Bottou & Bengio (1995); Song & Wang (2005); Hughes & Sudderth (2013); Pinto & Engel (2015). Typically, these are designed for fast and incremental learning of clusters without having to recompute clustering over the entire dataset. Despite presenting a similar online clustering algorithm, our goal is to jointly learn both online clusters and input representations that facilitate future online clustering episodes.

**Few-shot learning.** Our model can recognize new classes with only one or a few examples. Our prototype-based memory is also inspired by the Prototypical Network and its variants (Snell et al., 2017; Allen et al., 2019; Ren et al., 2021). Few-shot methods can reduce or remove reliance on class labels using semi- and self-supervised learning (Ren et al., 2018; Huang et al., 2019; Hsu et al., 2019; Gidaris et al., 2019; Antoniou & Storkey, 2019; Khodadadeh et al., 2019; Medina et al., 2020).

Classical few-shot learning, however, relies on episodes of equal number of training and test examples from a fixed number of new classes. Gidaris & Komodakis (2018); Triantafillou et al. (2020); Tao et al. (2020); Zhu et al. (2021) consider extending the standard episodes with incremental learning and varying number of examples and classes. Ren et al. (2021) proposed a new setup that incrementally accumulates new classes and re-visits old classes over a sequence of inputs. We evaluate our algorithm on a similar setup; however, unlike that work, our proposed algorithm does not rely on any class labels.

**Human category learning.** Our work is related to human learning settings and online clustering models from cognitive science (Carpenter & Grossberg, 1987; Fisher et al., 1991; Anderson, 1991; Love et al., 2004; Murphy, 2004; Lake et al., 2009). These models assume a known, fixed representation of inputs. In contrast, our model learns both representations and categories in an end-to-end fashion.

## 3 Online Unsupervised Prototypical Networks

We now introduce our model, *online unsupervised prototypical networks* (*OUPN*), which operates in a streaming categorization setting. At each time step $t$, OUPN receives an input $\mathbf{x}_t$ and predicts both a categorical variable $\hat{y}_t$ that indicates the object class and also a binary variable $\hat{u}_t$ that indicates whether the class is known ($u = 0$) or new ($u = 1$). OUPN uses a network $h$ to encode the input to obtain embedding $\mathbf{z}_t = h(\mathbf{x}_t; \theta)$, where $\theta$ represents the learnable parameters of the encoder network.

We first describe the inference procedure to cluster embeddings obtained by a fixed $\theta$ using an online probabilistic mixture model. Next, we propose a multi-component loss for representation learning in our setting which allows $\theta$ to be learned from scratch in the course of online clustering.

### 3.1 Inference

We formulate our clustering inference procedure in terms of a probabilistic mixture model, where each cluster corresponds to a Gaussian distribution $f(\mathbf{z}; \mathbf{p}, \sigma^2)$, with mean $\mathbf{p}$, a constant isotropic variance $\sigma^2$ shared across all clusters, and mixture weights $w$: $p(\mathbf{z}; P) = \sum_k w_k f(\mathbf{z}; \mathbf{p}_k, \sigma^2)$. Throughout a sequence, the number of components evolves as the model makes an online decision of when to create a new cluster or remove an old one. We assume that the prior distribution for the Bernoulli variable $u$ is constant—$u_0 \equiv \Pr(u = 1)$)—and the prior for a new cluster is uniform over the entire space—$z_0 \equiv \Pr(\mathbf{z}|u = 1)$ (Lathuilière et al., 2018). In the following, we characterize inference as an approximate extension of the EM algorithm to a streaming setting. The full derivation is included in Appendix A.

#### 3.1.1 E-step

Upon seeing the current input $\mathbf{z}_t$, the online clustering procedure needs to predict the cluster assignment or initiate a new cluster in the E-step.

**Inferring cluster assignments.** The categorical variable $\hat{y}$ infers the cluster assignment of the current input example with regard to the existing clusters. $\hat{y}_{t,k} = \Pr(y_t = k | \mathbf{z}_t, u = 0) = \frac{\Pr(\mathbf{z}_t|y_t=k,u=0)\Pr(y_t=k)}{\Pr(\mathbf{z}_t,u=0)} = \frac{w_k f(\mathbf{z}_t; \mathbf{p}_{t,k}, \sigma^2)}{\sum_{k'} w_{k'} f(\mathbf{z}_t; \mathbf{p}_{t,k'}, \sigma^2)} = \mathrm{softmax}\left(\log w_k - \frac{1}{\tau} d(\mathbf{z}_t, \mathbf{p}_{t,k})\right)$, where $w_k$ is the mixing coefficient of cluster $k$, $d(\cdot, \cdot)$ is the distance function, and $\tau$ is an independent learnable temperature parameter that is related to the cluster variance.

**Inference on unknown classes.** The binary variable $\hat{u}$ estimates the probability that the current input belongs to a new cluster: $\hat{u}_t = \Pr(u_t = 1 | \mathbf{z}_t) \geq \mathrm{sigmoid}((\min_k \frac{1}{\tau} d(\mathbf{z}_t, \mathbf{p}_{t,k}) - \beta)/\gamma)$, where $\beta$ and $\gamma$ are separate learnable parameters related to $z_0$ and $u_0$, allowing us to predict different confidence levels for unknown and known classes.

#### 3.1.2 M-step

Here we infer the posterior distribution of the cluster centroids $\Pr(\mathbf{p}_{t,k}|\mathbf{z}_{1:t})$. We formulate an efficient recursive online update, similar to Kalman filtering, incorporating the evidence of the current input $\mathbf{z}_t$ and avoiding re-clustering the entire input history. We define $\hat{\mathbf{p}}_{t,k}$ as the posterior estimate of the mean of the $k$-th cluster at time step $t$, and $\hat{c}_{t,k}$ is the estimate of the inverse variance.

**Updating centroids.** Suppose that in the E-step we have determined that $y_t = k$. Then the posterior distribution of the $k$-th cluster after observing $\mathbf{z}_t$ is:

$$\Pr(\mathbf{p}_{t,k}|\mathbf{z}_{1:t}, y_t = k) \propto \Pr(\mathbf{z}_t|\mathbf{p}_{t,k}, y_t = k)\Pr(\mathbf{p}_{t,k}|\mathbf{z}_{1:t-1})$$

$$\approx f(\mathbf{z}_t; \mathbf{p}_{t,k}, \sigma^2) \int_{\mathbf{p}'} f(\mathbf{p}_{t,k}; \mathbf{p}', \sigma_{t,d}^2) f(\mathbf{p}'; \hat{\mathbf{p}}_{t-1,k}, \hat{\sigma}_{t-1,k}^2)$$

$$= f(\mathbf{z}_t; \mathbf{p}_{t,k}, \sigma^2) f(\mathbf{p}_{t,k}; \hat{\mathbf{p}}_{t-1,k}, \sigma_{t,d}^2 + \hat{\sigma}_{t-1,k}^2).$$

The transition probability distribution $\Pr(\mathbf{p}_{t,k}|\mathbf{p}_{t-1,k})$ is a zero-mean Gaussian with variance $\hat{\sigma}_{t,d}^2 = (1/\rho - 1)\hat{\sigma}_{t-1,k}^2$, where $\rho \in (0,1]$ is some constant that we define to be the memory decay coefficient. Since the representations are learnable, we assume that $\sigma^2 = 1$, and the memory update equation can be formulated as follows: $\hat{c}_{t,k} = \mathbb{E}_{y_t}[\hat{c}_{t,k}|y_t] = \rho\hat{c}_{t-1,k} + \hat{y}_{t,k}(1 - \hat{u}_{t,k}); \hat{\mathbf{p}}_{t,k} = \mathbb{E}_{y_t}[\hat{\mathbf{p}}_{t,k}|y_t] = \mathbf{z}_t \frac{\hat{y}_{t,k}(1-\hat{u}_{t,k})}{\rho\hat{c}_{t-1,k}+1} + \hat{\mathbf{p}}_{t-1,k}\left(1 - \frac{\hat{y}_{t,k}(1-\hat{u}_{t,k})}{\rho\hat{c}_{t-1,k}+1}\right); \hat{w}_{t,k} = \mathbb{E}_{y_t}[\hat{w}_{t,k}|y_t] = \hat{c}_{t,k}/\sum_l \hat{c}_{t,l}$, where $\hat{c} \equiv 1/\hat{\sigma}_{t,k}^2$, which can be viewed a count variable for the number of elements in each estimated cluster, subject to the decay factor $\rho$ over time.

**Adding and removing clusters.** At any point in time, the mixture model is described by a collection of tuples $(\hat{\mathbf{p}}_k, \hat{c}_k)$. We convert the probability of whether an observation belongs to a new cluster into a decision: if $\hat{u}_t$ exceeds a threshold $\alpha$, we create a new cluster. Due to the decay factor $\rho$, our $\hat{c}$ estimate of a cluster can decay to zero over time, which is appropriate for modeling nonstationary environments. In practice, we keep a maximum number of $K$ clusters, and once the limit is reached, we simply pop out the weakest $\mathbf{p}_{k'}$, where $k' = \arg\min(\hat{w}_k)$: $P_t = P_{t-1} \setminus \{(\hat{\mathbf{p}}_{k'}, \hat{c}_{k'})\} \cup \{(\mathbf{z}_t, 1)\}$.

**Relation to Online ProtoNet.** The formulation of our streaming EM-like algorithm is similar to the Online ProtoNet (Ren et al., 2021), with several key differences. First, to handle nonstationary mixtures, we incorporate a decay term which is related to the variance of the transition probability. Second, our new cluster creation is unsupervised, whereas in (Ren et al., 2021), only labeled examples lead to new clusters. Third, representation learning in (Ren et al., 2021) relies on a supervised loss, whereas our objective—described in the next section—is entirely unsupervised. Nonetheless, to indicate the lineage of our model, OUPN, we refer to the cluster centroids as *prototypes* and the mixture model as a *prototype memory*.

## 3.2 Learning

A primary goal of our learning algorithm is to learn good visual representations through this online clustering process. We start the learning from scratch: the encoder network is randomly initialized, and the prototype memory will produce more accurate class predictions as the representations become more informative throughout learning. Our overall representation learning objective has three terms: $\mathcal{L} = \mathcal{L}_{\text{self}} + \lambda_{\text{ent}}\mathcal{L}_{\text{ent}} + \lambda_{\text{new}}\mathcal{L}_{\text{new}}$. This loss function drives the learning of the main network parameters $\theta$, as well as other learnable control parameters $\beta$, $\gamma$, and $\tau$. We explain each term in detail below.

1. **Self-supervised loss ($\mathcal{L}_{\text{self}}$):** Inspired by recent self-supervised representation learning approaches, we apply augmentations on $\mathbf{x}_t$, and encourage the clustering assignments to match across different views. Self-supervision follows three steps: First, the model makes a prediction on the augmented view, and obtains $\hat{y}$ and $\hat{u}$ (E-step). Secondly, it updates the prototype memory according to the prediction (M-step). To create a learning target, we query the original view again, and obtain $\tilde{y}$ to supervise the cluster assignment of the augmented view, $\hat{y}'$, as in distillation (Hinton et al., 2015). $\mathcal{L}_{\text{self}} = \frac{1}{T}\sum_t -\tilde{y}_t \log \hat{y}_t'$. Note that both $\tilde{y}_t$ and $\hat{y}_t'$ are produced after the M-step so we can exclude the "unknown" class in the representation learning objective. We here introduce a separate temperature parameter $\tilde{\tau}$ to control the entropy of the mixture assignment $\tilde{y}_t$.

2. **Entropy loss ($\mathcal{L}_{\text{ent}}$):** In order to encourage more confident predictions we introduce a loss function $\mathcal{L}_{\text{ent}}$ that controls the entropy of the original prediction $\hat{y}$, produced in the initial E-step: $\mathcal{L}_{\text{ent}} = \frac{1}{T}\sum_t -\hat{y}_t \log \hat{y}_t$.

3. **New cluster loss ($\mathcal{L}_{\text{new}}$):** Lastly, our learning formulation also includes a loss for initiating new clusters $\mathcal{L}_{\text{new}}$. We define it to be a Beta prior on the expected $\hat{u}$, and we introduce a hyperparameter $\mu$ to control the expected number of clusters: $\mathcal{L}_{\text{new}} = -\log\Pr(\mathbb{E}[\hat{u}])$. This acts as a regularizer on the total number of prototypes: if the system is too aggressive in creating prototypes, then it does not learn to merge instances of the same class; if it is too conservative, the representations can collapse to a trivial solution.

While there are several hyperparameters involved in inference and learning, in our experiments we only optimize a few: the Beta mean $\mu$, the threshold $\alpha$, the memory decay $\rho$, and the two loss term coefficients. The others are set to default values for all datasets and experiments. See Appendix B for a complete discussion of hyperparameters.

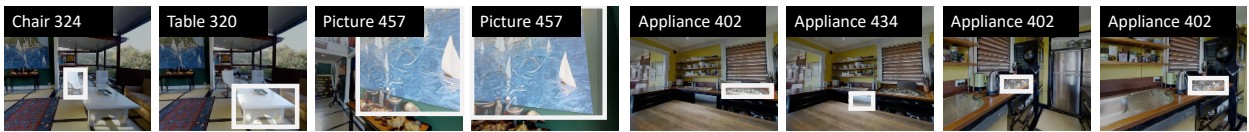

Figure 2: An example subsequence of the RoamingRooms dataset (Ren et al., 2021), consisting of glimpses of an agent roaming in an indoor environment, and the task is to recognize object instances.

**Full algorithm.** Let $\Theta = \{\theta, \beta, \gamma, \tau\}$ denote the union of the learnable parameters. Algorithm 1 outlines our proposed learning algorithm. The full list of hyperparameters are included in Appendix B.

---

**Algorithm 1** Online Unsupervised Prototypical Learning

---

**repeat**
    $\mathcal{L}_{\text{self}} \leftarrow 0$, $p_{\text{new}} \leftarrow 0$.
    **for** $t \leftarrow 1 \dots T$ **do**
        Observe new input $\mathbf{x}_t$.
        Encode input, $\mathbf{z}_t \leftarrow h(\mathbf{x}_t; \theta)$.
        Compare to existing prototypes: $[\hat{u}_t, \hat{y}_t] \leftarrow \text{E-step}(\mathbf{z}_t, P; \beta, \gamma, \tau)$.
        **if** $\hat{u}_t^0 < \alpha$ **then**
            Assign $\mathbf{z}_t$ to existing prototypes: $P \leftarrow \text{M-step}(\mathbf{z}_t, P, \hat{u}_t, \hat{y}_t)$.
        **else**
            Recycle the least used prototype if $P$ is full.
            Create a new prototype $P \leftarrow P \cup \{(\mathbf{z}_t, 1)\}$.
        **end if**
        Compute pseudo-labels: $[\_, \tilde{y}_t] \leftarrow \text{E-step}(\mathbf{z}_t, P; \beta, \gamma, \tilde{\tau})$.
        Augment a view: $\mathbf{x}'_t \leftarrow \text{augment}(\mathbf{x}_t)$.
        Encode the augmented view: $\mathbf{z}'_t \leftarrow h(\mathbf{x}'_t; \theta)$.
        Compare the augmented view to existing prototypes: $[\_, \hat{y}'_t] \leftarrow \text{E-step}(\mathbf{z}'_t, P; \beta, \gamma, \tau)$.
        Compute the self-supervision loss: $\mathcal{L}_{\text{self}} \leftarrow \mathcal{L}_{\text{self}} - \frac{1}{T} \tilde{y}_t \log \hat{y}'_t$.
        Compute the entropy loss: $\mathcal{L}_{\text{ent}} \leftarrow \mathcal{L}_{\text{ent}} - \frac{1}{T} \hat{y}_t \log \hat{y}_t$.
        Compute the average probability of creating new prototypes, $p_{\text{new}} \leftarrow p_{\text{new}} + \frac{1}{T} \hat{u}_t$.
    **end for**
    Compute the new cluster loss: $\mathcal{L}_{\text{new}} \leftarrow -\log \text{Pr}(p_{\text{new}})$.
    Sum up losses: $\mathcal{L} \leftarrow \mathcal{L}_{\text{self}} + \lambda_{\text{ent}} \mathcal{L}_{\text{ent}} + \lambda_{\text{new}} \mathcal{L}_{\text{new}}$.
    Update parameters: $\Theta \leftarrow \text{optimize}(\mathcal{L}, \Theta)$.
**until** convergence
**return** $\Theta$

---

It is worth noting that if we create a new prototype every time step, then OUPN is similar to a standard contrastive learning with an instance-based InfoNCE loss (Chen et al., 2020a; He et al., 2020); therefore it can be viewed as a generalization of this approach. Additionally, all the losses can be computed online without having to store any examples beyond the collection of prototypes.

**Practical implementation.** In practice, we make the following implementation choices. First, we use cosine similarity instead of negative squared Euclidean distance for computing the mixture logits, because cosine similarity is bounded and is found to be more stable to train. Second, when we perform cluster inference, we treat the mixing coefficients $w_k$ as constant and uniform as otherwise we find that the representations may collapse into a single large cluster.

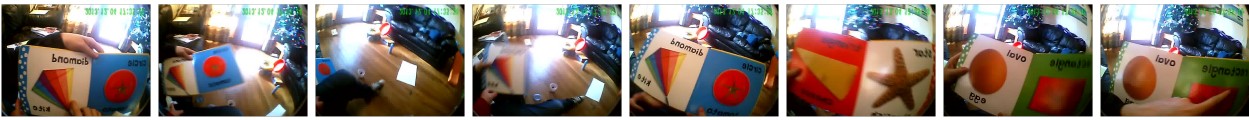

Figure 3: An example subsequence of the SAYCam dataset (Sullivan et al., 2022), consisting of egocentric videos collected from human babies.

## 4 Experiments

In this section, we evaluate our proposed learning algorithm on a set of visual learning tasks and examine the quality of the output categories. Contrasting with prior work on visual representation learning, our primary scenario of interest is online training with non-iid image sequences.

**Online clustering evaluation.** During evaluation we present our model a sequence of all new images (unlabeled or labeled) and we would like to see how well it produces a successful grouping of novel inputs. The class label index starts from zero for each sequence, and the classes do not overlap with the training set. The model memory is reset at the beginning of each sequence.

In unsupervised readout, the model directly predicts the label for each image, i.e. the model $g$ directly predicts $\hat{y}_t = g(\mathbf{x}_{1:t})$. In supervised readout (*for evaluation only*), the model has access to all labels up to time step $t-1$, and needs to predict the label for the $t$-th image, i.e. $\hat{y}_t = g(\mathbf{x}_{1:t}, y_{1:t-1})$. We used the following metrics to evaluate the quality of the grouping of test sequences:

- **Adjusted mutual information (AMI):** In the *unsupervised* setting, we use the mutual information metric to evaluate the similarity between our prediction $\{\hat{y}_1, \ldots, \hat{y}_T\}$ the groundtruth class ID $\{y_1, \ldots, y_T\}$. Since the online greedy clustering method admits a threshold parameter $\alpha$ to control the number of output clusters, therefore for each model we sweep the value of $\alpha$ to maximize the AMI score, to make the score threshold-invariant: $\text{AMI}_{\max} = \max_\alpha \text{AMI}(y, \hat{y}(\alpha))$. The maximization of $\alpha$ can be thought of as part of the readout procedure, and it is designed to particularly help other self-supervised learning baselines since their feature similarity functions are not necessarily calibrated for clustering.

- **Average precision (AP):** In the *supervised* setting, we followed the evaluation procedure in Ren et al. (2021) and used average precision, which combines both accuracy for predicting known classes as well as unknown ones.

**Offline readout evaluation.** A popular protocol to evaluate self-supervised representation learning is to use a classifier trained offline on top of the representations to perform semantic class readout. Because AMI and AP are designed to evaluate novel instance classification, we included offline evaluation protocols for semantic classes. We considered the following classifiers:

- **Nearest neighbor readout:** A common protocol is to use a k-nearest-neighbor classifier to readout the learned representations. For RoamingRooms we set $k = 39$ and for SAYCam we set $k = 1$, selected based on validation performance. The difference in $k$ is likely because in SAYCam the training clips overlap with test clips, whereas in RoamingRooms there is a larger difference between training and testing environments.

- **Linear readout:** Another popular protocol is to train a linear classifier on top of the learned representations to a given set of semantic classes. For RoamingRooms, we used the Adam optimizer with learning rate $10^{-3}$ for 20 epochs, and for SAYCam, we used the SGD optimizer with learning rate searched among {1.0, 0.1, 0.01} for each model for 100 epochs.

|  | AMI | AP | Acc. (k-NN,%) | Acc. (Linear,%) |
|---|---|---|---|---|
| **Supervised** | | | | |
| Supervised CNN | - | - | 72.11 | 71.93 |
| Online ProtoNet (Ren et al., 2021) | 79.02 | 89.94 | - | - |
| **Unsupervised** | | | | |
| Random Network | 28.25 | 11.68 | 28.84 | 26.73 |
| SimCLR (Chen et al., 2020a) | 50.03 | 52.98 | 44.84 | 48.83 |
| MoCo-V2 (Chen et al., 2020b) | 50.39 | 64.98 | 46.81 | 49.02 |
| SwAV (Caron et al., 2020) | 42.70 | 37.31 | 40.04 | 45.77 |
| SwAV+Queue (Caron et al., 2020) | 48.31 | 50.40 | 43.63 | 45.31 |
| SimSiam (Chen & He, 2021) | 47.58 | 44.15 | 43.99 | 48.24 |
| OUPN (Ours) | **78.16** | **84.86** | **48.37** | **52.28** |

Table 1: Instance and semantic class recognition results on RoamingRooms

| Random split | Acc. 1-NN | Linear |
|---|---|---|
| ImageNet | 72.67 | 53.23 |
| TC-S (Orhan et al., 2020) (iid) | 80.76 | 62.36 |
| Random | 10.04 | 9.37 |
| SimSiam | 26.53 | 19.91 |
| SwAV | 34.48 | 31.99 |
| SimCLR | 49.13 | 37.23 |
| MoCo-V2 | 53.45 | 36.12 |
| OUPN (Ours) | **64.35** | **44.29** |

| Subsample 10x | Acc. 1-NN | Linear |
|---|---|---|
| ImageNet | 48.09 | 40.61 |
| TC-S (Orhan et al., 2020) (iid) | 60.43 | 50.16 |
| Random | 9.74 | 15.15 |
| SimSiam | 18.71 | 17.39 |
| SwAV | 21.90 | 19.89 |
| SimCLR | 28.24 | 25.98 |
| MoCo-V2 | 30.18 | 25.97 |
| OUPN (Ours) | **36.52** | **30.25** |

Table 2: Semantic classification results on SAYCam (Child S)

**Competitive methods.** Our focus is online unsupervised visual representation learning. There are very few existing methods developed for this setting. To the best of our knowledge, continual unsupervised learning (Rao et al., 2019) (CURL) is the only directly comparable work, but this method relies on input reconstruction and scales poorly to more general environments. We include the comparison to CURL in the Appendix (Table 8). Unsupervised few-shot learning approaches are also related (Khodadadeh et al., 2019; Medina et al., 2020), but these methods are directly related to standard self-supervised learning methods. Therefore we compare OUPN with the following competitive self-supervised visual representation learning methods.

- **SimCLR** (Chen et al., 2020a) is a contrastive learning method with an instance-based objective that tries to classify an image instance among other augmented views of the same batch of instances. It relies on a large batch size and is often trained on well-curated datasets such as ImageNet (Deng et al., 2009).

- **MoCo** (He et al., 2020) is a contrastive learning method with an instance-based objective. It maintains a large representation queue to provide negative samples. The parameters of the queue encoder are momentum averaged. We have a similar mechanism as MoCo by having a queue of representations; however, we store and accumulate prototypes rather than individual frames. In this paper, we implement the V2 version of the algorithm (Chen et al., 2020b), and the queue size is set to 2400.

- **SwAV** (Caron et al., 2020) is a contrastive learning method with a clustering-based objective. It has a stronger performance than SimCLR on ImageNet. The clustering is achieved through Sinkhorn-Knopp which assumes balanced assignment, and prototypes are learned by gradient descent.

- **SwAV+Queue** is a SwAV variant with an additional example queue. This setup is proposed in Caron et al. (2020) to deal with small training batches. A feature queue that accumulates instances across batches allows the clustering process to access more data points. The queue size is set to 2000.

- **SimSiam** (Chen & He, 2021) is a self-supervised learning method that does not require negative samples. It uses a stop-gradient mechanism and a predictor network to make sure the representations do not collapse. Through not using negative samples, SimSiam could be immune to treating images of the same instances as negative samples.

For fair comparison on online representation learning, all of the above methods are trained on the *same* dataset using the same input data as our model, instead of using their pretrained checkpoints from ImageNet.

Since none of these competitive methods are designed to output classes with a few examples, we need an additional clustering-based readout procedure to compute AMI and AP scores. We use a simple online greedy clustering procedure for these methods. For each timestep, it searches for the closest prototype; in unsupervised mode, if it fails with $\hat{u}_t$ greater than $\alpha$, it will create a new prototype, and otherwise it will

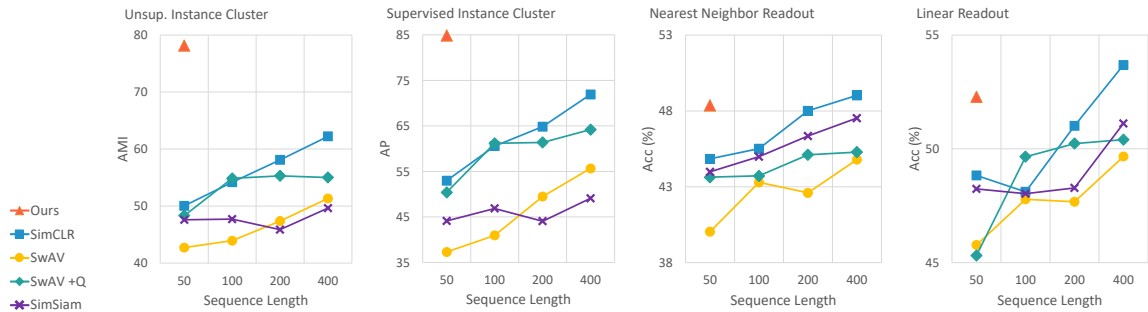

Figure 4: Comparison to SimCLR, SwAV, and SimSiam with larger batch sizes on RoamingRooms

aggregate the current embedding to the cluster centroid. As explained above, the $\alpha$ parameter is maximized for each model on its test scores to optimize performance.

## 4.1 Indoor home environments

We first evaluate the algorithm using the RoamingRooms dataset (Ren et al., 2021) where the images are collected from indoor environments (Chang et al., 2017) using a random walking agent. The dataset contains 1.22M image frames and 7K instance classes from 6.9K random walk episodes. The input for each frame is the RGB values and a object segmentation mask (in the 4th channel); the output (used here only for evaluation with AP and AMI) is the object instance ID. An example episode is shown in Fig. 2. The dataset is split into different home environments (60 training, 10 val, and 20 test). Each training iteration consists of a sequence of images from one of the homes. At test time, for the instance classification task, we ask the model to recognize novel objects in a new sequence of images in one of the test homes. For the semantic classification task, we ask the model to classify among 21 semantic categories including "picture", "chair", "lighting", "cushion", "table", etc.

SimCLR, SwAV and SimSiam use varying batch sizes (50, 100, 200, and 400). For online (non-iid) settings, the notion of batch size can be understood as "sequence length". Other training parameters can be found in the Appendix. Note that all baselines use the same training inputs as our model.

**Results.** Our main results are shown in Table 1. Although self-supervised methods, such as SimCLR, SwAV and SimSiam, have shown promising results on large batch learning on ImageNet, their performance here are relatively weak compared to the supervised baseline. In contrast, our method OUPN shows impressive performance on this benchmark: it almost matches the supervised learner in AMI, and reached almost 95% of the performance of the supervised learner in AP. OUPN also outperforms other methods in terms of k-NN and linear readout accuracy. We hypothesize that the nonstationary distribution of online frames could impose several challenges to standard self-supervised learning methods. First, SimCLR could treat adjacent similar frames as negative pairs. Second, it breaks SwAV's assumption on balanced cluster assignment and stationary cluster centroids. Adding a queue slightly improves SwAV; however, since the examples in the queue cannot be used to compute gradients, the nonstationary distribution still hampers gradient updates. Lastly, all of them could suffer from a very small batch size in our online setting.

To illustrate the impact of our small batch episodes, we increase the batch size for SimCLR and SwAV, from 50 to 400, at the cost of using multiple GPUs training in parallel. The results are shown in Fig. 4. Results indicate that increasing the batch size can improve these baselines, which matches our expectation. Nevertheless, our method using a batch size of 50 is still able to outperform other self-supervised methods using a batch size of 400, which takes $8\times$ computational resource compared to ours. Note that the large batch experiments are designed to provide the best setting for other self-supervised methods to succeed. We do not need to run our model with larger batch size since our prototype memory is a sequential module, and keeping the batch size smaller allows quicker online adaptation and less memory consumption.

**Comparison to iid modes of SimCLR, SwAV, and SimSiam.** The original SimCLR, SwAV, and SimSiam were designed to train on iid data. To study the effects of this assumption, we implemented an approx-

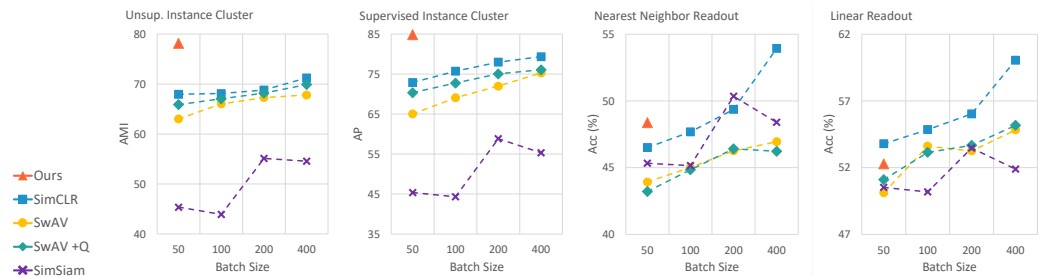

Figure 5: Comparison to iid-trained versions of SimCLR, SwAV, and SimSiam on RoamingRooms.

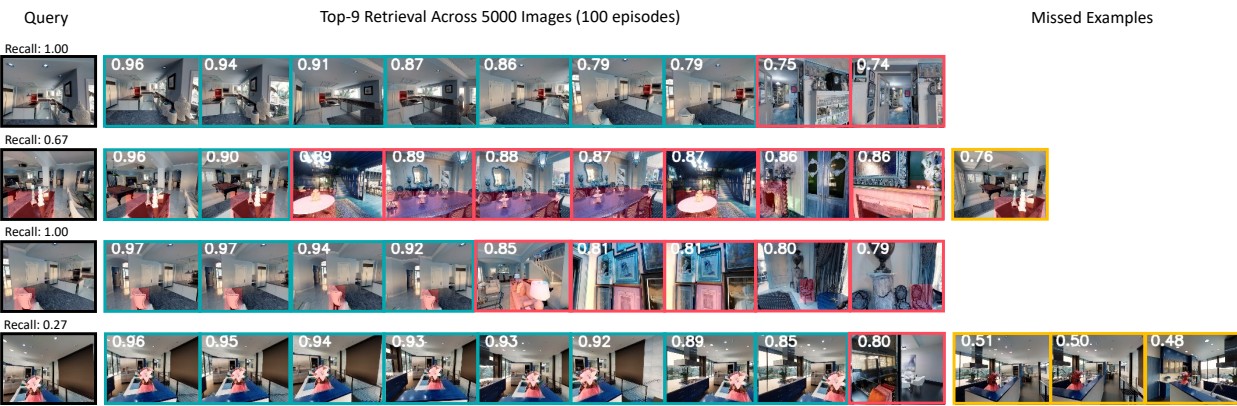

Figure 6: Image retrieval results on RoamingRooms. In each row, the leftmost image is the query image, and retrieved images are shown to its right. Cosine similarity scores on the top left; a green border denotes a correct retrieval, red false positive, and yellow a miss. Recall is the proportion of instances in the top-9.

imation to an iid distribution by using a large random queue that shuffles the image frames. As in the study shown in Fig. 5, we again vary the batch size for these competitive methods. All of these self-supervised baselines thrive with iid data; the gains of iid over non-iid can be seen by comparing Fig. 5 to Fig. 4. Larger batches help both methods again here. Interestingly, our method using a batch size of 50 non-iid data again outperforms both methods using a batch size of 400 of iid data in terms of AMI and AP. The only case where our method is inferior to SimCLR is when SimCLR is trained with large batches under iid setting on semantic classification readout. This is reasonable since semantic classification and iid large batch training is the setting SimCLR was originally developed for. Again, iid large batch training is not what we aim to solve in this paper, and we include the iid experiments in the paper simply to better understand the failure points of existing algorithms.

**Visualization on image retrieval.** To verify the usefulness of the learned representation, we ran an image retrieval visualization using the first 5000 images in the first 100 test sequences of length 50 and perform retrieval in a leave-one-out procedure. This procedure is only to visualize the similarity and is distinct from our evaluation procedure that requires class-label prediction.The results are shown in Fig. 6. Similarity scores are also provided. The top retrieved images are all from the same instance of the query image, and our model sometimes achieves perfect recall. This confirms that our model can handle a certain amount of view angle change. We also investigated the missed examples and we found that these are taken from more distinct view angles.

## 4.2 Head mounted camera recordings

Inspired by how humans acquire visual understanding ability after birth, we further evaluated our method on realistic first-person videos collected using baby egocentric cameras. The SAYCam dataset (Sullivan et al., 2022) is collected using 500 hours video data from three children. We obtained permission to use from the original authors. Following prior work (Orhan et al., 2020), we focused on using the Child S subset

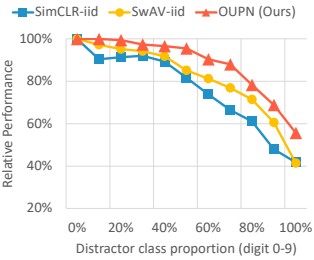 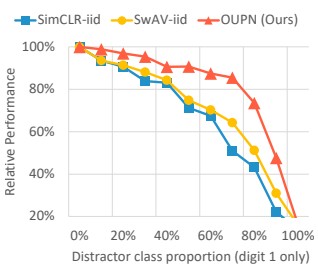

Figure 7: Robustness to imbalanced distributions by adding distractors (Omniglot mixed with MNIST images). Performance is relative to the original and a random baseline.

in our work. See Figure 3 for an example subsequence. We used MobileNet-V2 for this experiment. We sampled the video at 4 seconds per frame to form a temporal window of 5 minutes (75 images) for each mini-batch. The inputs are cropped and reshaped into $224 \times 224$ RGB images. We repeat the full 164-hour video for 16 times (16 epochs) for a total of 2624 hours for all methods trained on this dataset.

To evaluate the learned representations, Orhan et al. (2020) used a labeled dataset of the images containing 26 semantic classes such as *ball*, *basket*, *car*, *chair*, etc. Following their settings, we used two different splits of the dataset: a random iid split and a subsampled split, which was proposed to reduce the proportion of redundant images. We used both a linear and a nearest neighbor readout.

**Results.** Results are shown in Table 2. We are able to outperform competitive self-supervised learning methods. We also reproduced the performance of the temporal classification (TC) model (Orhan et al., 2020) and an ImageNet pretrained model for comparison. Since the TC model is trained using random iid samples of the full video, therefore it is understandable that our online streaming model performs worse. We also note that nearest neighbor readout generally performs better than linear readout on this benchmark, likely due to the existence of many similar frames in the video.

### 4.3 Handwritten characters and ImageNet images

We also evaluated our method on two different tasks: recognizing novel handwritten characters from Omniglot (Lake et al., 2015) and novel ImageNet classes. Here, images are static and are not organized in a video-like sequence, and models have to reason more about conceptual similarity between images to learn grouping. Furthermore, since this is a more controllable setup, we can test our hypothesis concerning sensitivity to class imbalance by performing manipulations on the episode distribution.

Our episodes are sampled from the RoamingOmniglot and RoamingImageNet dataset (Ren et al., 2021). An episode involves several different *contexts*, each consisting of a set of classes, and in each context, classes are sampled from a Chinese restaurant process. We use 150-frame episodes with 5 contexts for RoamingOmniglot and 48-frame with 3 contexts for RoamingImageNet.

|  | RoamingOmniglot | | RoamingImageNet | |
| --- | --- | --- | --- | --- |
|  | AMI | AP | AMI | AP |
| **Supervised** | | | | |
| Pretrain-Supervised | 84.48 | 93.83 | 29.44 | 24.39 |
| Online ProtoNet (Ren et al., 2021) | 89.64 | 92.58 | 29.73 | 25.38 |
| **Unsupervised** | | | | |
| Random Network | 17.66 | 17.01 | 4.55 | 2.65 |
| SimCLR (Chen et al., 2020a) | 59.06 | 73.50 | 6.87 | 12.25 |
| MoCo-V2 (Chen et al., 2020b) | 50.38 | 65.17 | 7.00 | 12.25 |
| SwAV (Caron et al., 2020) | 62.09 | 75.93 | 9.87 | 5.23 |
| SwAV+Queue (Caron et al., 2020) | 67.25 | 81.96 | 10.61 | 4.83 |
| SimSiam (Chen & He, 2021) | 45.57 | 56.12 | 12.64 | 6.31 |
| OUPN (Ours) | **84.42** | **92.84** | **19.03** | **15.05** |

Table 3: RoamingOmniglot and RoamingImageNet results

**Results.** The results are reported in Table 3. In both datasets, our method outperforms self-supervised baselines using the same batch size setting. In RoamingOmniglot, our model is able to significantly reduce the gap between supervised and unsupervised models, however in RoamingImageNet the gap is still wide, which suggests that our model is still less effective handling more distinct images of the same semantic class in the online stream.

**Effect of imbalanced distribution.** To achieve a better understanding of why OUPN performs better than other instance- and clustering-based self-supervised learning methods, here we study the effect of

imbalanced cluster sizes by manipulating the class distribution in the training episodes. In the first setting, we randomly replace Omniglot images with MNIST digits, with probability from 0% to 100%. For example, at 50% rate, an MNIST digit is over 300 times more likely to appear compared to any Omniglot character class, so the episodes are composed of half frequent classes and half infrequent classes. In the second setting, we randomly replace Omniglot images with MNIST digit 1 images, which makes the imbalance even greater. We compared our method to SimCLR and SwAV in the iid setup, since this is the scenario they were designed for. Results of the two settings are shown in Fig. 7, and our method is shown to be more robust under imbalanced distribution than SimCLR and SwAV. Compared to clustering-based methods like SwAV, our prototypes can be dynamically created and updated with no constraints on the number of elements per cluster. Compared to instance-based methods like SimCLR, our prototypes sample the contrastive pairs more equally in terms of representation similarity. We hypothesize that these model aspects contribute to the differences in robustness.

### 4.4 Ablation studies and hyperparameter optimization

Ablation studies on the terms in the objective function, as well as explorations of the effect of hyperparameter values, including the prototype memory size $K$, decay rate $\rho$, threshold $\alpha$, and Beta mean $\mu$, can be found in Appendix C.2.

## 5 Conclusion

Our goal is to develop learning procedures for real-world agents who operate online and in structured, nonstationary environments. Toward this goal, we develop an online unsupervised algorithm for discovering visual representations and categories. Unlike standard self-supervised learning, our algorithm embeds category formation in a probabilistic clustering module that is jointly learned with the representation encoder. Our clustering is more flexible and supports learning of new categories with very few examples. At the same time, we leverage self-supervised learning to acquire semantically meaningful representations. Our method is evaluated in both synthetic and realistic image sequences and it outperforms state-of-the-art self-supervised learning algorithms for both the non-iid sequences we are interested in as well as sequences transformed to be iid to better match assumptions of the learning algorithms.

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

# A  Method Derivation

## A.1  E-step

**Inferring cluster assignments.**  The categorical variable $\hat{y}$ infers the cluster assignment of the current input example with regard to the existing clusters.

$$\hat{y}_{t,k} = \Pr(y_t = k | \mathbf{z}_t, u = 0) \tag{1}$$

$$= \frac{\Pr(\mathbf{z}_t | y_t = k, u = 0) \Pr(y_t = k)}{\Pr(\mathbf{z}_t, u = 0)} \tag{2}$$

$$= \frac{w_k f(\mathbf{z}_t; \mathbf{p}_{t,k}, \sigma^2)}{\sum_{k'} w_{k'} f(\mathbf{z}_t; \mathbf{p}_{t,k'}, \sigma^2)} \tag{3}$$

$$= \frac{\exp(\log w_k - d(\mathbf{z}_t, \mathbf{p}_{t,k})/2\sigma^2)}{\sum_{k'} \exp(\log w'_k - d(\mathbf{z}_t, \mathbf{p}_{t,k'})/2\sigma^2)} \tag{4}$$

$$= \mathrm{softmax}\left(\log w_k - d(\mathbf{z}_t, \mathbf{p}_{t,k})/\tau\right), \tag{5}$$

$$= \mathrm{softmax}(v_{t,k}), \tag{6}$$

where $w_k$ is the mixing coefficient of cluster $k$ and $d(\cdot, \cdot)$ is the distance function, and $v_{t,k}$ is the logits. In our experiments, $w_k$'s are kept as constant and $\tau$ is an independent learnable parameter.

**Inference on unknown classes.**  The binary variable $\hat{u}$ estimates the probability that the current input belongs to a new cluster:

$$\hat{u}_t = \Pr(u_t = 1 | \mathbf{z}_t) \tag{7}$$

$$= \frac{z_0 u_0}{z_0 u_0 + \sum_k w_k f(\mathbf{z}_t; \mathbf{p}_{t,k}, \sigma^2)(1 - u_0)} \tag{8}$$

$$= \frac{1}{1 + \frac{1}{z_0 u_0} \sum_k w_k f(\mathbf{z}_t; \mathbf{p}_{t,k}, \sigma^2)(1 - u_0)} \tag{9}$$

$$= \frac{1}{1 + \exp(\log(\frac{1}{z_0 u_0} \sum_k w_k f(\mathbf{z}_t; \mathbf{p}_{t,k}, \sigma^2)(1 - u_0)))} \tag{10}$$

$$= \frac{1}{1 + \exp(-\log(z_0) - \log(u_0) + \log(1 - u_0) + \log(\sum_k w_k f(\mathbf{z}_t; \mathbf{p}_{t,k}, \sigma^2)))} \tag{11}$$

$$= \frac{1}{1 + \exp(-s + \log(\sum_k \exp(\log(w_k) - d(\mathbf{z}_t, \mathbf{p}_{t,k})/2\sigma^2)))} \tag{12}$$

$$= \mathrm{sigmoid}(s - \log(\sum_k \exp(\log(w_k) - d(\mathbf{z}_t, \mathbf{p}_{t,k})/2\sigma^2))) \tag{13}$$

$$= \mathrm{sigmoid}(s - \log(\sum_k \exp(v_{t,k}))), \tag{14}$$

where $s = \log(z_0) + \log(u_0) - \log(1 - u_0) + m\log(\sigma) + m\log(2\pi)/2$ and $m$ is the input dimension. In our implementation, we use max here instead of logsumexp since we found max leads to better and more stable training performance empirically. It can be derived as a lower bound:

$$\hat{u}_t = \mathrm{sigmoid}(s - \log(\sum_k \exp(\log(w_k) - d(\mathbf{z}_t, \mathbf{p}_{t,k})/2\sigma^2))) \tag{15}$$

$$\geq \mathrm{sigmoid}(s - \log(\max_k \exp(-d(\mathbf{z}_t, \mathbf{p}_{t,k})/2\sigma^2))) \tag{16}$$

$$= \mathrm{sigmoid}(s + \min_k d(\mathbf{z}_t, \mathbf{p}_{t,k})/2\sigma^2) \tag{17}$$

$$= \mathrm{sigmoid}((\min_k d(\mathbf{z}_t, \mathbf{p}_{t,k}) - \beta)/\gamma), \tag{18}$$

where $\beta = -2s\sigma^2$, $\gamma = 2\sigma^2$. To make learning more flexible, we directly make $\beta$ and $\gamma$ as independent learnable parameters so that we can control the confidence level for predicting unknown classes.

## A.2 M-step

Here we infer the posterior distribution of the prototypes $\Pr(\mathbf{p}_{t,k}|\mathbf{z}_{1:t})$. We formulate an efficient recursive online update, similar to Kalman filtering, by incorporating the evidence of the current input $\mathbf{z}_t$ and avoiding re-clustering the entire input history. We define $\hat{\mathbf{p}}_{t,k}$ as the estimate of the posterior mean of the $k$-th cluster at time step $t$, and $\hat{\sigma}_{t,k}^2$ is the estimate of the posterior variance.

**Updating prototypes.** Suppose that in the E-step we have determined that $y_t = k$. Then the posterior distribution of the $k$-th cluster after observing $\mathbf{z}_t$ is:

$$\Pr(\mathbf{p}_{t,k}|\mathbf{z}_{1:t}, y_t = k) \tag{19}$$

$$\propto \Pr(\mathbf{z}_t|\mathbf{p}_{t,k}, y_t = k)\Pr(\mathbf{p}_{t,k}|\mathbf{z}_{1:t-1}) \tag{20}$$

$$= \Pr(\mathbf{z}_t|\mathbf{p}_{t,k}, y_t = k)\int_{\mathbf{p}'}\Pr(\mathbf{p}_{t,k}|\mathbf{p}_{t-1,k} = \mathbf{p}')\Pr(\mathbf{p}_{t-1,k} = \mathbf{p}'|\mathbf{z}_{1:t-1}) \tag{21}$$

$$\approx f(\mathbf{z}_t; \mathbf{p}_{t,k}, \sigma^2)\int_{\mathbf{p}'} f(\mathbf{p}_{t,k}; \mathbf{p}', \sigma_{t,d}^2)f(\mathbf{p}'; \hat{\mathbf{p}}_{t-1,k}, \hat{\sigma}_{t-1,k}^2) \tag{22}$$

$$= f(\mathbf{z}_t; \mathbf{p}_{t,k}, \sigma^2)f(\mathbf{p}_{t,k}; \hat{\mathbf{p}}_{t-1,k}, \sigma_{t,d}^2 + \hat{\sigma}_{t-1,k}^2). \tag{23}$$

If we assume that the transition probability distribution $\Pr(\mathbf{p}_{t,k}|\mathbf{p}_{t-1,k})$ is a zero-mean Gaussian with variance $\sigma_{t,d}^2 = (1/\rho - 1)\hat{\sigma}_{t-1,k}^2$, where $\rho \in (0, 1]$ is some constant that we defined to be the memory decay coefficient, then the posterior estimates are:

$$\hat{\mathbf{p}}_{t,k}|_{y_t=k} = \frac{\mathbf{z}_t\hat{\sigma}_{t-1,k}^2/\rho + \hat{\mathbf{p}}_{t-1,k}\sigma^2}{\sigma^2 + \hat{\sigma}_{t-1,k}^2/\rho}, \quad \hat{\sigma}_{t,k}^2|_{y_t=k} = \frac{\sigma^2\hat{\sigma}_{t-1,k}^2/\rho}{\sigma^2 + \hat{\sigma}_{t-1,k}^2/\rho}. \tag{24}$$

If $\sigma^2 = 1$, and $\hat{c}_{t,k} \equiv 1/\hat{\sigma}_{t,k}^2$, $\hat{c}_{t-1,k} \equiv 1/\hat{\sigma}_{t-1,k}^2$, it turns out we can formulate the update equation as follows, and $\hat{c}_{t,k}$ can be viewed as a count variable for the number of elements in each estimated cluster, subject to the decay factor $\rho$ over time:

$$\hat{c}_{t,k}|_{y_t=k} = \rho\hat{c}_{t-1,k} + 1, \tag{25}$$

$$\hat{\mathbf{p}}_{t,k}|_{y_t=k} = \mathbf{z}_t\frac{1}{\hat{c}_{t,k}|_{y_t=k}} + \hat{\mathbf{p}}_{t-1,k}\frac{\rho\hat{c}_{t-1,k}}{\hat{c}_{t,k}|_{y_t=k}}. \tag{26}$$

If $y_t \neq k$, then the prototype posterior distribution simply gets diffused at timestep $t$:

$$\Pr(\mathbf{p}_{t,k}|z_{1:t}, y_t \neq k) \approx f(\mathbf{p}_{t,k}; \hat{\mathbf{p}}_{t-1,k}, \hat{\sigma}_{t-1,k}^2/\rho) \tag{27}$$

$$\hat{c}_{t,k}|_{y_t\neq k} = \rho\hat{c}_{t-1,k}, \tag{28}$$

$$\hat{\mathbf{p}}_{t,k}|_{y_t\neq k} = \hat{\mathbf{p}}_{t-1,k}. \tag{29}$$

Finally, our posterior estimates at time $t$ are computed by taking the expectation over $y_t$:

$$\hat{c}_{t,k} = \mathbb{E}_{y_t}[\hat{c}_{t,k}|_{y_t}] \tag{30}$$

$$= \hat{c}_{t,k}|_{y_t=k}\Pr(y_t = k|\mathbf{z}_t) + \hat{c}_{t,k}|_{y_t\neq k}\Pr(y_t \neq k|\mathbf{z}_t) \tag{31}$$

$$= (\rho\hat{c}_{t-1,k} + 1)\hat{y}_{t,k}(1 - \hat{u}_{t,k}) + \rho\hat{c}_{t-1,k}(1 - \hat{y}_{t,k}(1 - \hat{u}_{t,k})), \tag{32}$$

$$= \rho\hat{c}_{t-1,k} + \hat{y}_{t,k}(1 - \hat{u}_{t,k}), \tag{33}$$

$$\hat{\mathbf{p}}_{t,k} = \mathbb{E}_{y_t}[\hat{\mathbf{p}}_{t,k}|_{y_t}] \tag{34}$$

$$= \hat{\mathbf{p}}_{t,k}|_{y_t=k}\Pr(y_t = k|\mathbf{z}_t) + \hat{\mathbf{p}}_{t,k}|_{y_t\neq k}\Pr(y_t \neq k|\mathbf{z}_t) \tag{35}$$

$$= \left(\mathbf{z}_t\frac{1}{\hat{c}_{t,k}|_{y_t=k}} + \hat{\mathbf{p}}_{t-1,k}\frac{\rho\hat{c}_{t-1,k}}{\hat{c}_{t,k}|_{y_t=k}}\right)\hat{y}_{t,k}(1 - \hat{u}_{t,k}) + \hat{\mathbf{p}}_{t-1,k}(1 - \hat{y}_{t,k}(1 - \hat{u}_{t,k})) \tag{36}$$

$$= \mathbf{z}_t\frac{\hat{y}_{t,k}(1 - \hat{u}_{t,k})}{\rho\hat{c}_{t-1,k} + 1} + \hat{\mathbf{p}}_{t-1,k}\left(1 - \hat{y}_{t,k}(1 - \hat{u}_{t,k}) + \hat{y}_{t,k}(1 - \hat{u}_{t,k})\frac{\rho\hat{c}_{t-1,k}}{\rho\hat{c}_{t-1,k} + 1}\right) \tag{37}$$

$$= \mathbf{z}_t\frac{\hat{y}_{t,k}(1 - \hat{u}_{t,k})}{\rho\hat{c}_{t-1,k} + 1} + \hat{\mathbf{p}}_{t-1,k}\left(1 - \frac{\hat{y}_{t,k}(1 - \hat{u}_{t,k})}{\rho\hat{c}_{t-1,k} + 1}\right). \tag{38}$$

Table 4: Experiment details for RoamingRooms

| Hyperparameter | Values |
|---|---|
| $\tau$ init | 0.1 |
| $\beta$ init | -12.0 |
| $\gamma$ init | 1.0 |
| Num prototypes $K$ | 150 |
| Memory decay $\rho$ | 0.995 |
| Beta mode $\mu$ | 0.5 |
| Entropy loss $\lambda_{\text{ent}}$ | 0.0 |
| New cluster loss $\lambda_{\text{new}}$ | 0.5 |
| Threshold $\alpha$ | 0.5 |
| Pseudo label temperature ratio $\tilde{\tau}/\tau$ | 0.1 |

Since $\hat{c}_{t,k}$ is also our estimate on the number of elements in each cluster, we can use it to estimate the mixture weights,

$$\hat{w}_{t,k} = \frac{\hat{c}_{t,k}}{\sum_{k'} \hat{c}_{t,k}}. \tag{39}$$

Note that in our experiments the mixture weights are not used and we assume that each cluster has an equal mixture probability.

## B Experiment Details

We provide additional implementation details in Tab. 4, 5, 6 and 7.

**RoamingRooms.** For baseline self-supervised learning methods, learning rate is scaled based on batch size $/256 \times 0.3$ using the default LARS optimizer with cosine learning rate decay and 1 epoch of linear learning rate warmup. We trained for a total of 10,240,000 examples. So the total number of training steps is 10,240,000 / batch size. For our proposed model, we used the batch size of 50 and trained for a total of 80,000 steps (4,000,000 examples), using the Adam optimizer and staircase learning rate decay starting from $10^{-3}$, with $10\times$ learning rate decay at 40k and 60k training steps. All data augmentation parameters are the same as the original SimCLR paper, except that in RoamingRooms the minimum crop area is changed to 0.2 instead of the default 0.08. Other details can be found in Table 4. We used a single Nvidia GeForce GTX 1080Ti GPU for running the standard experiments (with batch size 50). For larger batch size self-supervised learning experiments, we used up to 8 GPU in a data-parallel setup. On average, training of OUPN takes 1.28 episodes per second, and full training of 80,000 steps takes 17 hours using 8G GPU memory and 5G CPU memory. In comparison, SimCLR trains at a speed of 1.42 episodes per second and also 8G GPU memory, which suggests that our sequential memory does not add too much computation overhead. Evaluation speed is around 3.20 episodes per second.

**SAYCam.** Data augmentation is slightly different from the standard static image setting. We found that there were a lot of blurred and shaking frames in the videos. Therefore, we added random rotation, motion blur and Gaussian blur in the data augmentation procedure for all methods (including the baselines). Motion blur is generated with a uniformly random direction between $[0°, 360°)$, with the length to be 5% of the image height, and Gaussian blur is generated by a blur kernel of 5% of the image height with the standard deviation to be uniform between [0.1, 1.2). Same to all the baselines, our SAYCam model also applies two different augmentations on each image in the input pair.

For baseline self-supervised learning methods, the learning rate is scaled based on the batch size $/256 \times 0.3$, or 0.0879 (batch size = 75), using the default LARS optimizer with cosine learning rate decay and 1 epoch of linear learning rate warmup. We trained the models for a total of 16 epochs. The total number of training steps is 31,568 (1,973 steps per epoch). For the TC-S model, we used the Adam optimizer with learning rate

Table 5: Experiment details for SAYCam

| Hyperparameter | Values |
|---|---|
| Random motion blur | 30% probability |
| Random Gaussian blur | 20% probability |
| Random rotation | uniform between -15° and 15° |
| $\tau$ init | 0.1 |
| $\beta$ init | -12.0 |
| $\gamma$ init | 1.0 |
| Num prototypes $K$ | 75 |
| Memory decay $\rho$ | 0.99 |
| Beta mean $\mu$ | 0.6 (mode=0.7) |
| Entropy loss $\lambda_{\text{ent}}$ | 0.0 |
| New cluster loss $\lambda_{\text{new}}$ | 0.3 |
| Threshold $\alpha$ | 0.5 |
| Pseudo label temperature ratio $\tilde{\tau}/\tau$ | 0.0 (i.e. one-hot pseudo labels) |

1e-3 and batch size 75. We trained it for 38k steps with 10× learning rate decays at 25k and 35k. For our model, we used the Adam optimizer with learning rate 1e-3, for a total of 30k training steps, with a 10× learning rate decay at 20k steps.

For $\hat{y}$ and $\hat{u}$, we found it was helpful to sample binary values for the two variables in the forward pass, and use gradient straight-through estimator in the backward pass. This modification was only applied on SAYCam experiments. Other details can be found in Table 5.

We used a single Nvidia GeForce GTX 1080Ti GPU for running the standard experiments (with batch size 75). On average, training of OUPN takes around 2.60 seconds per episode, and full training of 30,000 steps takes under 22 hours using 11G GPU memory and 7.5G CPU memory. In comparison, SimCLR trains at a speed of 1.42 episodes per second and also 11G GPU memory, which suggests that our sequential memory does not add too much computation overhead. Evaluation speed is around 3.20 episodes per second.

**RoamingOmniglot and RoamingImageNet.** For baseline self-supervised learning methods on RoamingOmniglot, the learning rate is scaled based on the batch size $/256 \times 0.5$, or 0.293 (batch size = 150), using the default LARS optimizer with cosine learning rate decay and 10 epochs of linear learning rate warmup. We trained the model for a total of 1,000 epochs. The total number of training steps is 527,000 (527 per epoch).

For baseline self-supervised learning methods on RoamingImageNet, the learning rate is scaled based on the batch size $/256 \times 0.3$, or 0.05625 (batch size = 48), using the default LARS optimizer with cosine learning rate decay and 1 epoch of linear learning rate warmup. We trained the models for a total of 10 epochs. The total number of training steps is 93,480 (9,348 per epoch).

For our model on both datasets, we train using the Adam optimizer with learning rate 1e-3, for a total of 80k training steps, with 10× learning rate decay at 40k and 60k. More implementation details can be found in Table 6 and 7. We used a single Nvidia GeForce GTX 1080Ti GPU for running the experiments (with batch size 150 for RoamingOmniglot and 48 for RoamingImageNet).

## B.1 Metric Details

For each method, we used the same nearest centroid algorithm for online clustering. For unsupervised readout, at each timestep, if the closest centroid is within threshold $\alpha$, then we assign the new example to the cluster, otherwise we create a new cluster. For supervised readout, we assign examples based on the class label, and we create a new cluster if and only if the label is a new class. Both readout procedures will provide us a sequence of class IDs, and we will use the following metrics to compare our predicted class IDs and groundtruth class IDs. Both metrics are designed to be threshold invariant.

Table 6: Experiment details for RoamingOmniglot

| Hyperparameter | Values |
|---|---|
| $\tau$ init | 0.1 |
| $\beta$ init | -12.0 |
| $\gamma$ init | 1.0 |
| Num prototypes $K$ | 150 |
| Memory decay $\rho$ | 0.995 |
| Beta mean $\mu$ | 0.5 |
| Entropy loss $\lambda_{\text{ent}}$ | 1.0 |
| New cluster loss $\lambda_{\text{new}}$ | 1.0 |
| Threshold $\alpha$ | 0.5 |
| Pseudo label temperature ratio $\tilde{\tau}/\tau$ | 0.2 |

Table 7: Experiment details for RoamingImageNet

| Hyperparameter | Values |
|---|---|
| $\tau$ init | 0.1 |
| $\beta$ init | -12.0 |
| $\gamma$ init | 1.0 |
| Num prototypes $K$ | 600 |
| Memory decay $\rho$ | 0.99 |
| Beta mean $\mu$ | 0.5 |
| Entropy loss $\lambda_{\text{ent}}$ | 0.5 |
| New cluster loss $\lambda_{\text{new}}$ | 0.5 |
| Threshold $\alpha$ | 0.5 |
| Pseudo label temperature ratio $\tilde{\tau}/\tau$ | 0.0 (i.e. one-hot pseudo labels) |

**AMI.** For unsupervised evaluation, we consider the adjusted mutual information score. Suppose we have two clustering $U = \{U_i\}$ and $V = \{V_j\}$, and $U_i$ and $V_j$ are set of example IDs, and $N$ is the total number of examples. $U$ and $V$ can be viewed as discrete probability distribution over cluster IDs. Therefore, the mutual information score between $U$ and $V$ is:

$$\text{MI}(U, V) = \sum_{i=1}^{|U|} \sum_{j=1}^{|V|} \frac{|U_i \cap V_j|}{N} \log \left( \frac{N|U_i \cap V_j|}{|U_i||V_j|} \right) \tag{40}$$

$$= \sum_{i=1}^{R} \sum_{j=1}^{C} \frac{n_{ij}}{N} \log \left( \frac{N n_{ij}}{a_i b_j} \right). \tag{41}$$

The adjusted MI score[1] normalizes the range between 0 and 1, and subtracts the baseline from random chance:

$$\text{AMI}(U, V) = \frac{MI(U, V) - \mathbb{E}[MI(U, V)]}{\frac{1}{2}(H(U) + H(V)) - \mathbb{E}[MI(U, V)]}, \tag{42}$$

where $H(\cdot)$ denotes the entropy function, and $\mathbb{E}[MI(U, V)]$ is the expected mutual information by chance [2]. Finally, for each model, we sweep the threshold $\alpha$ to get a threshold invariant score:

$$\text{AMI}_{\max} = \max_{\alpha} \text{AMI}(y, \hat{y}(\alpha)). \tag{43}$$

**AP.** For supervised evaluation, we used the AP metric. The AP metric is also threshold invariant, and it takes both output $\hat{u}$ and $\hat{y}$ into account. First it sorts all the prediction based on its unknown score $\hat{u}$ in

---

[1]`https://scikit-learn.org/stable/modules/generated/sklearn.metrics.adjusted_mutual_info_score.html`
[2]`https://en.wikipedia.org/wiki/Adjusted_mutual_information`

Table 8: Unsupervised iid learning on Omniglot using an MLP

| Method | 3-NN Error | 5-NN Error | 10-NN Error |
|---|---|---|---|
| VAE (Joo et al., 2020) | 92.34±0.25 | 91.21±0.18 | 88.79±0.35 |
| SBVAE (Joo et al., 2020) | 86.90±0.82 | 85.10±0.89 | 82.96±0.64 |
| DirVAE (Joo et al., 2020) | 76.55±0.23 | 73.81±0.29 | 70.95±0.29 |
| CURL (Rao et al., 2019) | 78.18±0.47 | 75.41±0.34 | 72.51±0.46 |
| SimCLR (Chen et al., 2020a) | 44.35±0.55 | 42.99±0.55 | 44.93±0.55 |
| SwAV (Caron et al., 2020) | **42.66±0.55** | 42.08±0.55 | 44.78±0.55 |
| OUPN (Ours) | 43.75±0.55 | **42.13±0.55** | **43.88±0.55** |

Table 9: Effect of mem. size $K$

| $K$ | RoamingOmniglot | | RoamingRooms | |
|---|---|---|---|---|
| | AMI | AP | AMI | AP |
| 50 | 89.19 | 95.12 | 75.33 | 82.42 |
| 100 | **90.54** | 95.83 | 76.70 | 83.51 |
| 150 | 90.24 | **95.92** | 77.07 | 84.00 |
| 200 | 90.36 | 95.68 | 76.81 | **84.45** |
| 250 | 89.87 | 95.69 | **77.83** | 84.33 |

Table 10: Effect of decay rate $\rho$

| $\rho$ | RoamingOmniglot | | RoamingRooms | |
|---|---|---|---|---|
| | AMI | AP | AMI | AP |
| 0.9 | 51.12 | 64.19 | 65.07 | 75.50 |
| 0.95 | 79.78 | 89.30 | 74.33 | 81.92 |
| 0.99 | 89.43 | 95.54 | 76.97 | 84.05 |
| 0.995 | **90.80** | **95.90** | **77.78** | **85.02** |
| 0.999 | 86.27 | 93.69 | 38.89 | 39.37 |

Table 11: Effect of $\lambda_{\text{new}}$

| $\lambda_{\text{new}}$ | RoamingOmniglot | | RoamingRooms | |
|---|---|---|---|---|
| | AMI | AP | AMI | AP |
| 0.0 | 38.26 | 93.40 | 19.49 | 73.93 |
| 0.1 | 86.60 | 93.50 | 67.25 | 71.69 |
| 0.5 | 89.89 | 95.28 | **78.04** | **84.85** |
| 1.0 | **90.06** | **95.81** | 77.59 | 84.36 |
| 2.0 | 88.74 | 95.73 | 77.62 | 84.72 |

ascending order. Then it checks whether $\hat{y}$ makes the correct prediction. For the N top ranked instances in the sorted list, it computes: precision@N and recall@N among the known instances.

- precision@$N = \frac{1}{N} \sum_n \mathbb{1}[\hat{y}_n = y_n]$,

- recall@$N = \frac{1}{K} \sum_n \mathbb{1}[\hat{y}_n = y_n]$,

where $K$ is the true number of known instances among the top N instances. Finally, AP is computed as the area under the curve of (y=precision@N, x=recall@N). For more details, see Appendix A.3 of Ren et al. (2021).

## C    Additional Experimental Results

### C.1    Comparison to Reconstruction-Based Methods

We additionally provide Tab. 8 to show a comparison with CURL (Rao et al., 2019) in the iid setting. We used the same MLP architecture and applied it on the Omniglot dataset using the same data split. Reconstruction-based methods lag far behind self-supervised learning methods. Our method is on par with SimCLR and SwAV.

### C.2    Additional Studies on Hyperparameters

In Table 9, we investigate the effect of the size of the prototype memory, and whether the model would benefit from a larger memory. It turns out that as long as the size of the memory is larger than the length of the input sequence for each gradient update step, it can learn good representations and the size is not a major determining factor.

In Table 10, we examine whether the memory forgetting parameter is important to the model. We found that the forgetting rate between 0.99 and 0.995 is the best. 0.999 (closer to no forgetting) results in worse performance.

In Table 11, we investigate the effect of various values for the new cluster loss coefficient. The optimal value is between 0.5 and 1.0. Without the new cluster loss, the model may learn to collapse all representations together into a single cluster.

Table 12: Effect of threshold $\alpha$

| | RoamingOmniglot | | RoamingRooms | |
|---|---|---|---|---|
| $\alpha$ | AMI | AP | AMI | AP |
| 0.3 | 82.75 | 90.57 | 52.60 | 58.71 |
| 0.4 | 81.59 | 90.94 | 59.69 | 66.11 |
| 0.5 | **89.65** | **95.22** | **77.96** | **84.34** |
| 0.6 | 87.01 | 93.87 | 64.65 | 69.49 |
| 0.7 | 86.08 | 92.94 | 66.60 | 73.54 |

Table 13: Effect of $\tilde{\tau}$

| | RoamingOmniglot | | RoamingRooms | |
|---|---|---|---|---|
| $\tilde{\tau}$ / $\tau$ | AMI | AP | AMI | AP |
| 0.05 | 89.23 | 95.01 | 77.44 | 84.38 |
| 0.10 | 89.71 | 95.21 | **77.89** | **84.99** |
| 0.20 | **89.78** | **95.31** | 77.82 | 84.57 |
| 0.50 | 89.40 | 95.13 | 76.81 | 83.90 |
| 1.00 | 89.62 | 95.16 | 0.00 | 19.91 |

Table 14: Effect of $\lambda_{\text{ent}}$

| | RoamingOmniglot | | RoamingRooms | |
|---|---|---|---|---|
| $\lambda_{\text{ent}}$ | AMI | AP | AMI | AP |
| 0.00 | 82.45 | 90.66 | **76.64** | **84.11** |
| 0.25 | 87.31 | 93.85 | 76.61 | 83.16 |
| 0.50 | 87.98 | 94.21 | 75.46 | 81.78 |
| 0.75 | 88.77 | 94.74 | 74.76 | 79.91 |
| 1.00 | **89.70** | **95.14** | 75.32 | 80.29 |

Table 15: Effect of mean $\mu$ of the Beta prior

| | RoamingOmniglot | | RoamingRooms | |
|---|---|---|---|---|
| $\mu$ | AMI | AP | AMI | AP |
| 0.3 | 84.14 | 93.19 | 68.75 | 72.58 |
| 0.4 | 86.59 | 93.10 | 69.19 | 73.86 |
| 0.5 | **89.89** | **95.24** | **77.61** | **84.64** |
| 0.6 | 85.93 | 93.81 | 64.21 | 73.23 |
| 0.7 | 26.22 | 92.08 | 48.58 | 64.28 |

In Table 12, the threshold parameter is found to be the best at 0.5. However, this could be correlated with how frequently the frames are sampled.

In Table 13, we found that the soft distillation loss is beneficial and slightly improves the performance compared to hard distillation because it may preserve some level of uncertainty in the pseudo labels.

In Table 14, the entropy loss we introduced leads to a significant improvement on the Omniglot dataset but not on the RoamingRooms dataset. This is likely because Omniglot is an easier task, so having the entropy loss helps the model boost its confidence, whereas in RoamingRooms, increasing model confidence may not be beneficial.

The Beta $\mu$ is computed as the following: Suppose $a$ and $b$ are the parameters of the Beta distribution, and $\mu$ is the mean. We fix $a = 4\mu$ and $b = 4 - a$. In Table 15, we found that the mean of the Beta prior is the best at 0.5. It has more impact on the RoamingRooms dataset, and has less impact on the RoamingOmniglot dataset. This parameter could be influenced by the total number of clusters in each sequence.

# D  Additional Visualization Results

We visualize the clustering mechanism and the learned image embeddings on RoamingRooms in Fig. 8 and 9. The results suggest that our model can handle a certain level of view point changes by grouping different view points of the same object into a single cluster. It also shows that our model is instance-sensitive: for example, the headboard, pillows, and the blanket are successfully separated.

In Fig. 10 and 11, we visualize the learned categories in RoamingOmniglot using t-SNE (Van der Maaten & Hinton, 2008). Different colors represent different ground-truth classes. Our method is able to learn meaningful embeddings and roughly group items of similar semantic meanings together.

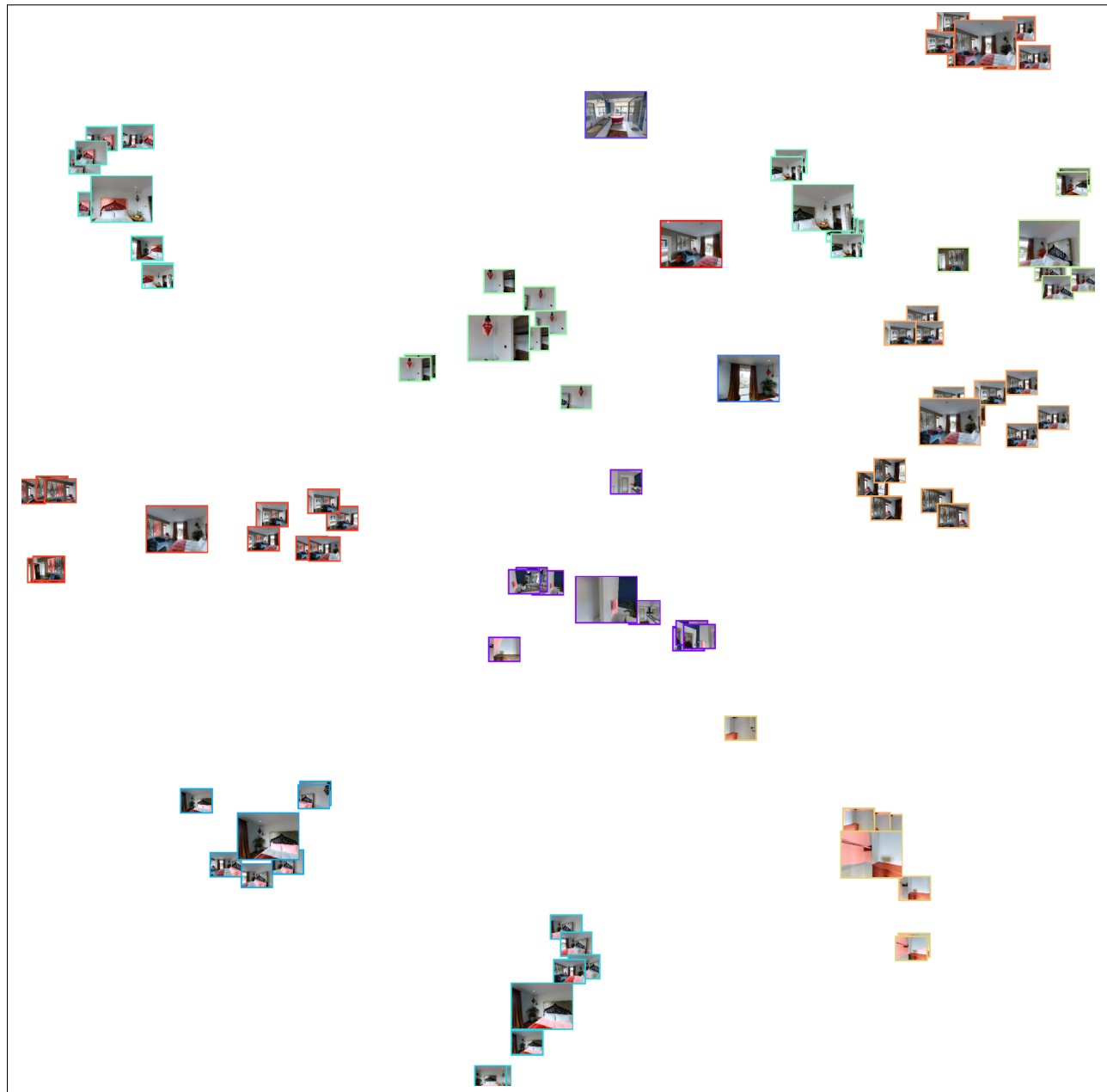

Figure 8: Embeddings and clustering outputs of an example episode (1). Embeddings are extracted from the trained CNN and projected to 2D space using t-SNE (Van der Maaten & Hinton, 2008). The main object in each image is highlighted in a red mask. The nearest example to each cluster centroid is enlarged. Image border colors indicate the cluster assignment.

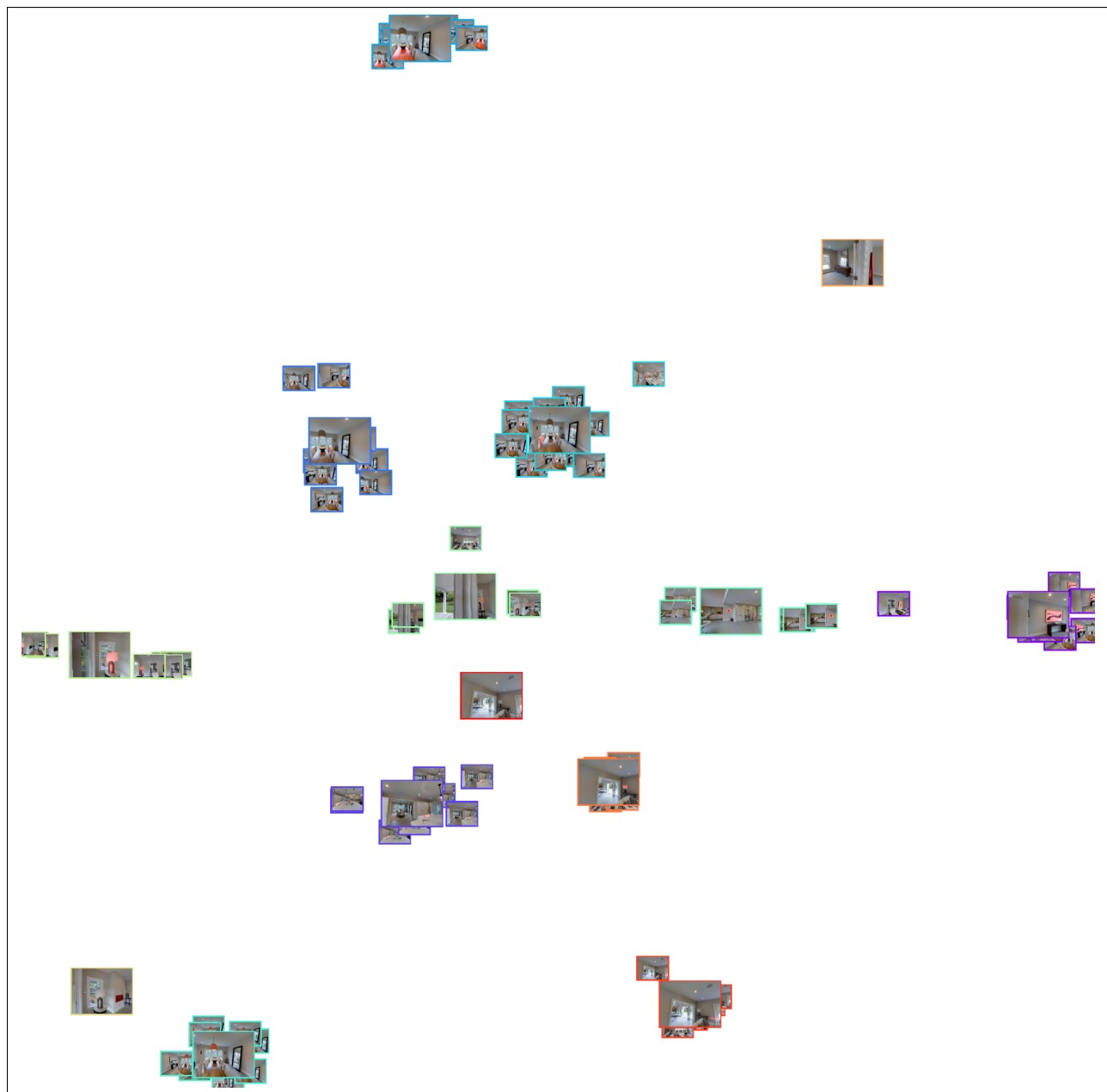

Figure 9: Embeddings and clustering outputs of another example episode (2).

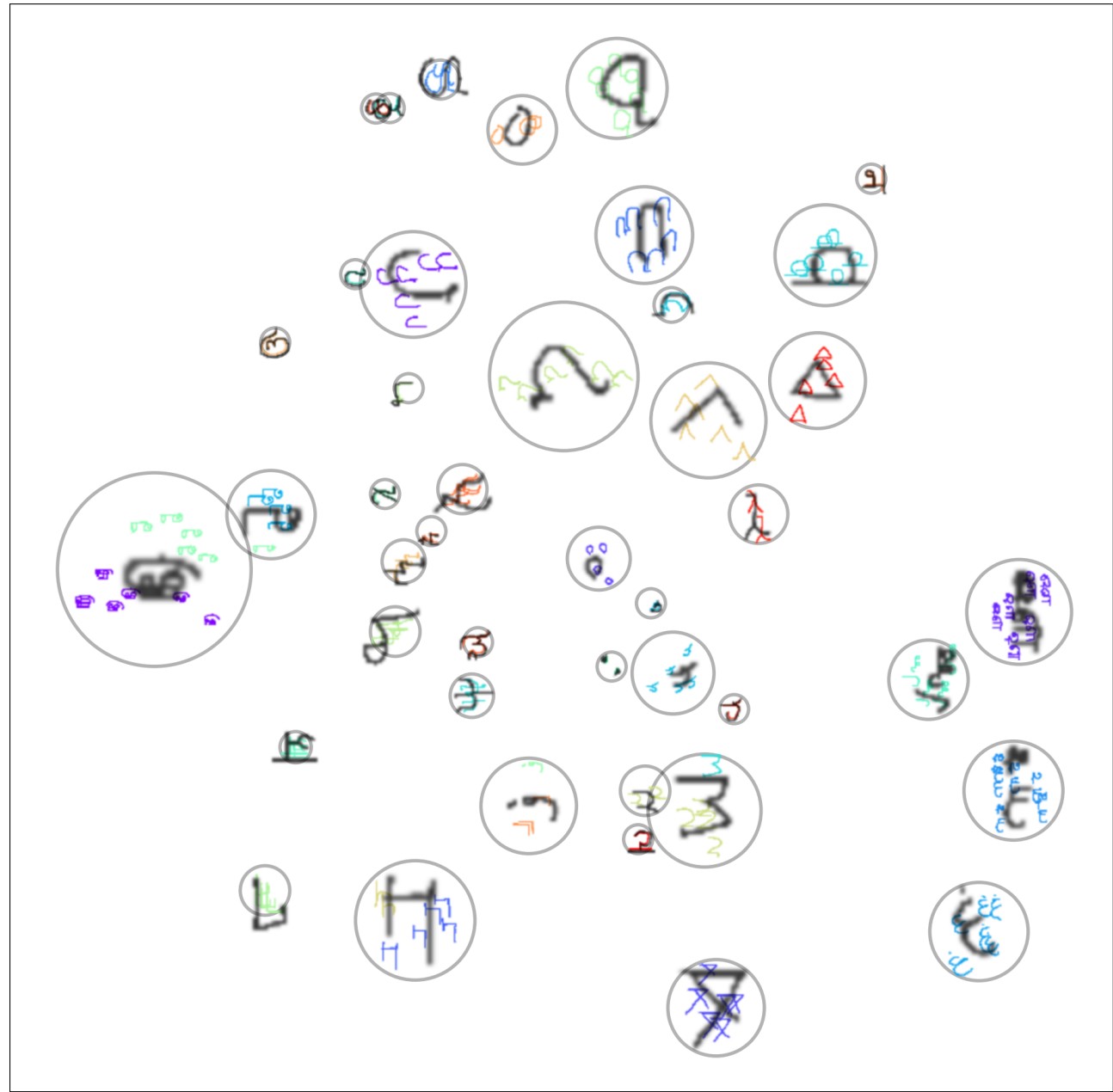

Figure 10: Embedding visualization of an unsupervised training episode of RoamingOmniglot. Different colors denote the ground-truth class IDs.

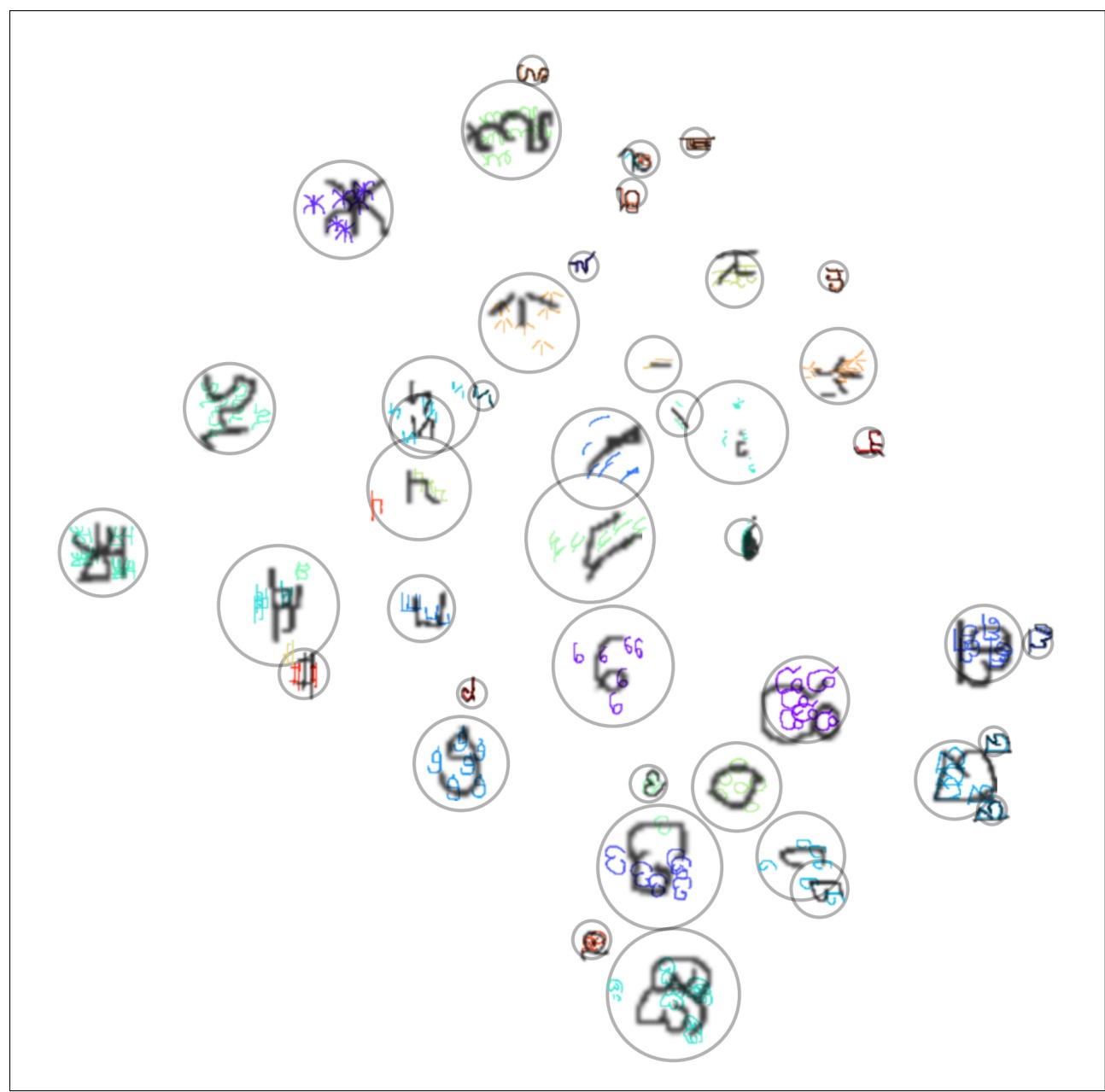

Figure 11: Embedding visualization of an test episode of RoamingOmniglot.

