# OpenReview forum: "Online Unsupervised Learning of Visual Representations and Categories"
_TMLR — Rejected by TMLR_

### Review · Reviewer_VJn3 · 2022-10-15

**Summary Of Contributions:**

The submitted manuscript presents a novel approach to unsupervised online representation learning, i.e. on continuous streams of data that is based on a changing distribution of classes which, in particular, may also be unknown. Learning a model that is able to adapt to the changing data distribution is formulated as an EM optimization problem. To this end, pseudo labels are inferred by an online clustering algorithm while gradually updating the corresponding prototypes, i.e adding and removing prototypes to reflect the current estimate of the data distribution. Comparison with established contrastive learning methods for unsupervised offline learning indicates the superiority of the presented method for learning from a changing data distribution.

**Audience:**

Yes

**Broader Impact Concerns:**

I have no concerns regarding the broader impact of the presented work.

**Claims And Evidence:**

Yes

**Requested Changes:**

To recommend the presented manuscript for acceptance, I would like to see the experiments I already mentioned in the weaknesses section:
- A comparison to CURL in the main results, i.e. Tab.1 and Tab.2, as it seem to be an appropriate baseline coming from the 'Continual Learning' community.
- Conducting the experiments in Fig. 4 and Fig. 5 also on RoamingImageNet, to have a more solid ablation study based on a more diverse data distribution.

**Strengths And Weaknesses:**

#### Strengths:

- The paper is very well written and is easy to follow.
- Formulating the task of online representation learning from a changing data distribution by means of an expectation maximization (EM) algorithm seems to be sound and effective.
- Based on the presented experiments the proposed approach proves to be superior than offline unsupervised methods when learning from changing data distributions.

#### Weaknesses:
- Well-known lines of research on online representation learning problem is 'Incremental Learning' or 'Continual Learning'. The related work CURL (Rao et al.) on Continual Learning described as 'To the best of our knowledge, continual unsupervised learning (Rao et al., 2019) (CURL) is the only directly comparable work'. However, there seem to be very few comparisons to CURL in the experimental section. While comparison are completely missing in the main results, only in the appendix C.1., Tab. 8, a comparison is presented. However, the setting of this experiment, i.e. Unsupervised *iid* learning on Omniglot using an MLP, does not seem to reflect the actual problem tackled in this manuscript.

- There seem to be 10 method-specific hyperparameters for adjusting the presented work (e.g. see appendix) to new datasets which may affect the robustness of the proposed approach

- Fig. 4 and 5 indicate that when adapting the learning setting, i.e. increasing the batch sizes and i.i.d. settings, established SSL approaches like SimCLR and SwaV may perform equally well in the non-i.i.d episode setting and even better when trained in an i.i.d. setting. In would be interesting to see similar experiments on RoamingImageNet which has a more diverse data distribution.

---

> ### Author Response · Authors · 2022-11-15
> **Response to reviewer VJn3**
>
> Thank you very much for your comments.
>
> > **There seem to be very few comparisons to CURL in the experimental section. While comparison are completely missing in the main results, only in the appendix C.1., Tab. 8, a comparison is presented. However, the setting of this experiment, i.e. Unsupervised iid learning on Omniglot using an MLP, does not seem to reflect the actual problem tackled in this manuscript.**
>
> Thank you for your suggestion. We would like to clarify why we do not think CURL is a good comparison to our approach.
>
> 1. CURL mainly uses a particular data stream where it first learns one class for tens of thousands of steps, and then another, in an incremental fashion. Although the algorithm is unsupervised, the model still implicitly assumes that a mini-batch has the same digit (for MNIST), or alphabet label (for Omniglot). By contrast, RoamingOmniglot is a continual few-shot benchmark, so alphabets switch much faster—it only stays in one alphabet for a few images. It is therefore much harder to tell when there is an alphabet switch in RoamingOmniglot. In this regard, RoamingOmniglot is actually closer to an “iid” setting for CURL since the alphabet switching happens within a mini-batch.
>
> 2. The CURL algorithm is based on VAE and its learning objective is based on data reconstruction. It is known to only work on small scale binary images such as MNIST or Omniglot. It is non-trivial to adapt the algorithm to larger RGB images, and the original authors did not attempt to try any experiments on that. In Table 8, we have already shown that the self-supervised learning learns much better image representations than CURL on Omniglot. Therefore, it is expected that CURL will perform much worse than self-supervised learning on larger RGB images. It would be interesting, however, to combine CURL in the latent space but such an upgrade will be a major effort that is outside the scope of our paper.
>
> Based on the reasons above, we think CURL is likely not a competitive method, especially on larger RGB images. We hope this addresses your concern. If not, we are happy to hear your further comments.
>
> > **There seem to be 10 method-specific hyperparameters for adjusting the presented work (e.g. see appendix) to new datasets which may affect the robustness of the proposed approach.**
>
> We did introduce a few hyperparameters, but we found that the amount of hyperparameter tuning was not too burdensome when we transferred our method to different datasets. As shown in Table 9-15, only the decay rate $\rho$ and the new cluster loss $\lambda_{\text{new}}$ have a noticeable impact on the performance, whereas the threshold $\alpha$ and Beta mean $\mu$ are optimal around 0.5. We have added a few more lines of explanation in the Appendix C.2.
>
> > **Fig. 4 and 5 indicate that when adapting the learning setting, i.e. increasing the batch sizes and i.i.d. settings, established SSL approaches like SimCLR and SwaV may perform equally well in the non-i.i.d episode setting and even better when trained in an i.i.d. setting. I would be interesting to see similar experiments on RoamingImageNet which has a more diverse data distribution.**
>
> Thanks for the suggestion. We have considered the setting; however RoamingImageNet is only a synthetic dataset, and the batches come from the original episodes from Ren et al. (2021) (Wandering Within a World). In order to change the batch size, we will have to change other sampling parameters, e.g. number of environments, number of classes per environment, probability of new classes, switching probability, etc. Unlike RoamingRooms, it is more complicated to vary the size of the batch in RoamingImageNet. We did try with sequence length to be 150 images, and the difference between our model and the best SSL method was reduced to 17.04% vs. 16.45%. In iid settings, our OUPN does worse than large batch SSL. This is likely due to incorrect clustering, since the clustering module of our model assumes that the adjacent images are likely to have similar class labels.

---

> > ### Comment · Reviewer_VJn3 · 2022-11-19
> > **Thank you for the response.**
> >
> > Thank you very much for your response.
> >
> > Re. Comparison with CURL approach:
> > After having read your response, I agree that CURL might not be a good comparison to your approach and is consequently not required to be included in the final manuscript.
> >
> > Re. my initial request for additional experiments with i.i.d. SSL approaches, i.e. "Fig. 4 and 5 indicate that when adapting the learning setting, i.e. increasing the batch sizes and i.i.d. settings, established SSL approaches like SimCLR and SwaV may perform equally well in the non-i.i.d episode setting and even better when trained in an i.i.d. setting. I would be interesting to see similar experiments on RoamingImageNet which has a more diverse data distribution.":
> >
> > While providing the requested experiments indeed seems to entail some effort to be made, e.g. adjusting the  "sampling parameters, e.g. number of environments, number of classes per environment, probability of new classes, switching probability", I still think it is doable without an inappropriate amount of effort. In particular I am convinced that these experiments will resolve open questions, thus add value to the overall quality of the experimental section. Hence, I still encourage the authors to conduct the suggested experiments.

---

### Review · Reviewer_4z4q · 2022-10-18

**Summary Of Contributions:**

This work aims to learn visual representations from real-world online data in an unsupervised manner, which cannot satisfy the i.i.d. data assumption in previous unsupervised learning approaches. To this end, this work proposes the online unsupervised prototypical network (OUPN), to learn visual representations and object categories on-the-fly, and adapt to imbalanced distributions of natural online data. OUPN learns visual representations as an online Gaussian mixture model, where the clusters can be dynamically adjusted. OUPN is evaluated on several benchmarks with online streaming data and imbalanced classes, and significantly outperforms previous unsupervised learning approaches including SimCLR, SwAV, and SimSiam.

**Audience:**

Yes

**Broader Impact Concerns:**

No concerns.

**Claims And Evidence:**

Yes

**Requested Changes:**

1. As described in the weaknesses, the ablation of OUPN designs could be improved.

2. As described in the weaknesses, the comparison of computation efficiency would be helpful.

3. Section 4, “Nearest neighbor readout”: Here drastically different $k$ are set for the two datasets. Could authors provide an explanation how $k$ is selected?

4. Minor issues:

    * Section 3.1.1, “Inference on unknown classes”: $\sigma$ should be referring to the sigmoid function, but it has appeared as the variance in Gaussian distribution (Section 3.1). Please change this notation.

    * Section 3.2: A missing link in “We include full details of our algorithm in Algorithm 1 in Appendix ??”


**Strengths And Weaknesses:**

Strengths:

1. This work proposes an unsupervised learning approach, OUPN, that can successfully learn from online data with imbalanced class distributions. Compared to previous approaches that assume i.i.d. training data and are applied to curated datasets like ImageNet, this approach lifts the limitation of unsupervised learning and shows how to learn from more realistic, online data streams.

2. The proposed approach OUPN is evaluated on several online datasets, including RoamingRooms, RoamingOmniglot, and RoamingImageNet, and shows significant advantage over previous unsupervised learning methods. Notably, OUPN outperforms previous methods even when they are provided with i.i.d. data samples and large batch sizes (Figures 4 and 5).

Weaknesses:

1. The ablation of OUPN designs needs to be improved:

    * Overall, OUPN learns both the representation and categories of visual objects. The visual representation is mainly learned through contrastive learning, and the categories are learned by a Gaussian mixture. It is unclear how the two parts interact with each other, and how each part improves upon previous unsupervised learning approaches. For example, the authors may evaluate OUPN’s learned representation in conjunction with the greedy clustering (as described in Section 4 for evaluating competitive methods for online clustering), to understand how the Gaussian mixture is beneficial to the overall approach.

   * The ablation study on hyperparameters (Appendix C.2) lacks sufficient explanation for readers to understand the effects of each component. As an example, why a separate, smaller $\tilde\tau$ is required for estimating the cluster of augmented views?

2. The maximum number of clusters, $K$, is pre-defined and fixed. In other words, the proposed approach still requires prior knowledge about the online data from humans, and cannot dynamically adjust the maximum number of clusters. Thus, it needs to anticipate how many categories to observe, or may learn visual representations that are not well separated from each other. This can be a limitation for applying this method to real-world data where the number of categories cannot be estimated in advance.

3. It would be better to show the computation efficiency of OUPN in comparison with previous unsupervised learning methods by listing the training throughput (image/second). As shown in Algorithm 1, multiple expectation steps are required in each iteration. Would they introduce too much computation overhead?

---

> ### Author Response · Authors · 2022-11-15
> **Response to reviewer 4z4q**
>
> Thank you very much for your review comments.
>
> > **It is unclear how the two parts interact with each other, and how each part improves upon previous unsupervised learning approaches. For example, the authors may evaluate OUPN’s learned representation in conjunction with the greedy clustering (as described in Section 4 for evaluating competitive methods for online clustering), to understand how the Gaussian mixture is beneficial to the overall approach.**
>
> Our AMI and AP scores are designed to exactly satisfy this request. We use the representations learned from each model, and use a greedy clustering on top to get the class labels. Since the greedy clustering is also part of OUPN, these scores directly evaluate how good the mixtures are. By contrast, linear readout and kNN readout are only evaluating the representations. Figures 8-11 in Appendix D also visualize the greedy clustering results.
>
> > **Why a separate, smaller $\tilde{tau}$ is required for estimating the cluster of augmented views?**
>
> A separate tau gives us another degree of freedom to control the pseudo label to be more discrete or soft. In some experiments, it is better to make the pseudo label soft, whereas in other experiments, it is better to make them discrete.
>
> > **The maximum number of clusters, K, is pre-defined and fixed. In other words, the proposed approach still requires prior knowledge about the online data from humans, and cannot dynamically adjust the maximum number of clusters. Thus, it needs to anticipate how many categories to observe, or may learn visual representations that are not well separated from each other. This can be a limitation for applying this method to real-world data where the number of categories cannot be estimated in advance.**
>
> Note that K is only fixed when computing the contrastive learning loss – we take the K most recent clusters to contrast with the current cluster. In fact, as we motivate throughout the paper, the clusters are being dynamically and incrementally created and can theoretically be kept in the memory forever. It is true that in the experiments we have kept K pre-defined, but we think it won’t affect the generality of the algorithm.
>
> > **It would be better to show the computation efficiency of OUPN in comparison with previous unsupervised learning methods by listing the training throughput (image/second). As shown in Algorithm 1, multiple expectation steps are required in each iteration. Would they introduce too much computation overhead?**
>
> Thanks for the suggestion. We have updated our manuscript and include these details in Appendix B. To summarize, the cost of computation is relatively small. We used a single GTX 1080Ti GPU for a standard experiment (the ones in the main tables). For experiments using larger batch sizes, up to 8 GPUs are used in parallel. The training speed is around 1.28 episodes per second (for RoamingRooms), and a single run can finish within 17 hours. In comparison, SimCLR is 1.47 episodes per second using a similar amount of GPU memory. So our method does not introduce too much overhead due to the sequential memory design.
>
> The expectation steps are basically performing a softmax operation across all prototypes. For a few hundred prototypes, they are not expensive operations.
>
> > **Section 4, “Nearest neighbor readout”: Here drastically different k are set for the two datasets. Could authors provide an explanation how k is selected?**
>
> We selected k between 1 to 39 for each model based on the validation performance. SAYCam models always prefer a smaller k because there are similar frames seen in the training set (due to the way how the dataset is originally split, there are overlaps between training and testing clip frames). By contrast, RoamingRooms models always prefer a larger k, since it is split by different home environments, which means there is a larger difference between training and testing, and it is better to average the prediction with a larger k. We have updated these details in the newer version.
>
> > **Minor issues in Section 3.1.1 & Section 3.2**
>
> Thank you for pointing them out. We have fixed both issues.

---

### Review · Reviewer_devn · 2022-11-01

**Summary Of Contributions:**

This paper advances a new strategy of self-supervised learning that conducts both visual representation learning and online few-shot learning of new object categories at the same time.
The authors argue that dominant approaches for self-supervised representation learning built upon the assumption that the unlabeled data is i.i.d. and well curated (e.g., ImageNet-like datasets) may have a hard time generalizing for more generic datasets with imbalanced classes and non-i.i.d. structure.
This if often the case for real-world dataset collected on the fly, e.g., virtual agents, babies discovering the world, that do not display the same curation and class balance as ImageNet.

The authors propose OUPN (Online Unsupervised Prototypical Network) to this effect. OUPN leverages prototype representations (previously used in few shot learning and recently in contrastive learning) to conduct the two desired functions: (1) learn to classify an input sample, using a Gaussian mixture over the prototypes, either by assigned it to the nearest prototype or by initiating a new prototype whenever the class of the input sample is novel, (2) learn visual representations thanks to a distillation loss between augmented views of the input.
For the clustering part the authors devise a sound and interesting formalism based on EM: (E) infer cluster assignments; (M) infer the posterior distributions of the prototypes using an online update inspired from Kalman filtering.

OUPN is tested on a number of datasets and settings (online clustering, offline evaluation - kNN classifier, linear probing, non-i.i.d. regimes for popular image datasets) against a series of relevant baselines from self-supervised learning, achieving interesting performance.

**Audience:**

Yes

**Broader Impact Concerns:**


No potential ethical concerns to signal.


**Claims And Evidence:**

Yes

**Requested Changes:**


Here are some changes that could be addressed in the updated work:

1) Clarification on the online clustering evaluation protocol and shift to Hungarian Matching

2) Addition of an analysis of the computational cost

3) Addition of stats for the data distribution of the considered datasets

4) Use of MoCo-V2 baseline

5) Expand related work with other non-i.i.d. attempts



**Strengths And Weaknesses:**


### Recommendation
I enjoyed reading and discovering this work and I think it proposes an interesting idea and corresponding strategy. Devising SSL strategies beyond the canonical ImageNet setting is certainly an interesting research avenue and the current OUPN method that does both representation learning and few shot learning for new categories is a improvements towards this. I'm leaning towards accepting this work for TMLR.

### Paper strengths

**Significance:**
- this paper proposes an interesting way of thinking of self-supervised learning (SSL) beyond the, now usual, ImageNet setting with curation and class balancing.
- with a few exceptions, SSL-based representation learning had not been much studied for different types of datasets with different specs
- the results obtained by OUPN are rather good.

**Clarity:**
- I find the paper mostly well written and argued. The intuition, reasoning and formalism are well explained (with some more details in the appendix)
- the authors conduct several sensitivity studies on the hyper-parameters of OUPN for better understanding

**Quality:**
- the approach seems techbically sound. There are several experiments on different settings and different relevant baselines considered.

**Originality:**
- I find this work quite quite original both in the offered perspective and in the implementation
- the authors cover a broad range of related works spanning self-supervisd learning, few-shot learning video-based representation learning, online learning, sequence modeling, etc.




### Paper weaknesses

#### Evaluation:
- there a few unclear or improvable aspects in the experiments part:
- for the online clustering evaluation it seems the authors use the greedy clustering assignment to map the prediction of the model (assignment to prototypes) to the ground truth classes. Given that the authors use a significantly higher number of prototypes than classes (150 vs 21 on RoamingRooms), with the greedy assignment the scores will be artificially increased, while not showing much practical utility since a class will be indicated by different prototypes
- I would have preferred a Hungarian matching strategy (often used in SSL for semantic segmentation [a], [b], [c]) that always maps a prototype to a single class

- for the k-NN experiment on RoamingRooms (Figure 6), is the instance mask concatenated to the RGB input?


#### Baselines
- not a major weakness or request, but from recent works studying SSL outside of ImageNet [d],[e], it appears that MoCo-V2 generalizes better than other approaches and it would be interesting to include it here, though I believe it will not outperform OUPN


#### Computational cost:
- except for the comparison of batch-sizes, there is no discussion on the computational complexity of this approach, e.g., cost a forward and backward pass in terms of time, compute, memory, etc., compared to a known SSL approach


#### Number of hyper-params
- the number of hyper-parameters introduced by this approach is relatively high, hinting a certain tuning that may be necessary for adapting to new settings
- the authors argue that only 5 of them (mean \mu, threshold \alpha, memory decay \rho and two loss term coefficients) need more thorough tuning


#### Data distribution
- since the non-i.i.d. character of the target datasets is addressed here, it would be useful to show the class distribution of the considered datasets and how imbalanced they are compared to Imagenet

#### Related work:
- there are few mildly related works from the SSL representation learning that attempt going beyond ImageNet or at least image-instance learning paradigm: targeting semantic segmentation [a-c], object detection [f], non-ImageNet datasets [d-e]


**References:**

[a] Cho et al., PiCIE: Unsupervised Semantic Segmentation using Invariance and Equivariance in Clustering, CVPR 2021

[b] Hamilton et al., Unsupervised Semantic Segmentation by Distilling Feature Correspondences, ICLR 2022

[c] Vobecky et al., Drive&Segment: Unsupervised Semantic Segmentation of Urban Scenes via Cross-modal Distillation, ECCV 2022

[d] Chen et al., MultiSiam: Self-supervised Multi-instance Siamese Representation Learning for Autonomous Driving, ICCV 2021

[e] Van Gansbeke et al., Revisiting Contrastive Methods for Unsupervised Learning of Visual Representations, NeurIPS 2021

[f] Henaff et al., Efficient Visual Pretraining with Contrastive Detection, ICCV 2021

---

> ### Author Response · Authors · 2022-11-15
> **Response to reviewer devn**
>
> Thank you very much for your review comments.
>
> > **Why is the number of prototypes in the model greater than classes?**
>
> The number of prototypes needs to be greater than the total classes since during training. Since the prototypes are not cleared after an episode and serves as a buffer, it could provide more negative choices for the contrastive learning objective.
>
> > **Why not use a Hungarian matching strategy?**
>
> We are not sure which part you suggest to use the Hungarian matching strategy.
>
> When performing online clustering, we must decide whether a new input belongs to an existing cluster or a new cluster, and if we decide to choose from the old clusters, then we match to the closest cluster. Since clustering is essentially a set of 1-to-many relationships, we think that the matching algorithm wouldn’t apply here (unlike evaluating segmentation or detection).
>
> In our evaluation of unsupervised clustering, we used AMI (adjusted mutual information), which is a popular metric measuring the closeness between two clusterings. In order to achieve a perfect score, you must have a 1-1 exact match.
>
> We hope our response clarifies your concern. If not, let us know and we are happy to revise the paper.
>
> > **For the k-NN experiment on RoamingRooms (Figure 6), is the instance mask concatenated to the RGB input?**
>
> Yes. For the k-NN experiment, the instance mask is part of the input as the fourth channel, and we extract the features from a learned encoder. And k-NN is applied on top of the extracted features to retrieve the semantic classes.
>
> > **It appears that MoCo-V2 generalizes better than other approaches and it would be interesting to include it here.**
>
> Thank you for the suggestion. We have added the MoCo-V2 for comparison in our newer version. It is sometimes slightly better than SimCLR, thanks to the representation queue buffer. However, as you pointed out, our method still performs better than MoCo-V2.
>
> > **Except for the comparison of batch-sizes, there is no discussion on the computational complexity of this approach, e.g., cost a forward and backward pass in terms of time, compute, memory, etc., compared to a known SSL approach.**
>
> Thank you for the suggestion. We have updated our manuscript and include these details in Appendix B. To summarize, the cost of computation is relatively small. We used a single GTX 1080Ti GPU for a standard experiment (the ones in the main tables). For experiments using larger batch sizes, up to 8 GPUs are used in parallel. The training speed is around 1.28 episodes per second (for RoamingRooms), and a single run can finish within 17 hours. In comparison, SimCLR is 1.47 episodes per second using a similar amount of GPU memory. So our method does not introduce too much overhead due to the sequential memory design.
>
> > **The number of hyper-parameters introduced by this approach is relatively high, hinting a certain tuning that may be necessary for adapting to new settings**
>
> We did introduce some hyperparameters, but we found that the amount of hyperparameter tuning was not too burdensome when we transferred our method to different datasets. As shown in Table 9-15, only the decay rate $\rho$ and the new cluster loss $\lambda_{\text{new}}$ have a noticeable impact on the performance, whereas the threshold $\alpha$ and Beta mean $\mu$ are optimal around 0.5. We have added a few more lines of explanation in Appendix C.2.
>
> > **Since the non-i.i.d. character of the target datasets is addressed here, it would be useful to show the class distribution of the considered datasets and how imbalanced they are compared to Imagenet.**
>
> Thank you for the suggestion. The class distribution of RoamingRooms can be found in Figure 3 from Ren et al. (2021) (https://arxiv.org/pdf/2007.04546.pdf). The class distribution of SAYCam can be found in Figure 1b of Orhan et al. (2020) (https://arxiv.org/pdf/2007.16189.pdf).
>
> > **There are few mildly related works from the SSL representation learning that attempt going beyond ImageNet or at least image-instance learning paradigm: targeting semantic segmentation [a-c], object detection [f], non-ImageNet datasets [d-e].**
>
> Thank you for the pointers. We have added them in the related work section of the newer version.

---

> > ### Comment · Reviewer_devn · 2022-11-30
> > **Clarification on Hungarian matching strategy**
> >
> > Thank you for the thorough responses.
> >
> > I would like to clarify the question regarding the Hungarian matching. As mentioned in my review, my concern was regarding the evaluation part, where it would be interesting to see how well to the clustering computed by OUPN follows the actual classes.
> > This will offer a somehow more clear view on the performance that can be obtained by this approach in such a setting where we constrain one cluster to match one single class in the evaluation.
> > I agree that to some extent the proposed AMI (adjusted mutual information) metric illustrates how close the two class distributions are overall, but it's not as easily to interpret as accuracy or mIoU. AMI is good, but it would be nice to be complemented by a simple additional metric.
> >
> > Thanks again. I don't have any more questions or comments at this point.

---

### Comment · Action_Editors · 2022-11-03
**Initiate A Discussion on Paper #472**

Dear authors and reviewers,

Please read all the reviews carefully and comprehensively. Then, we can start a discussion on the questions, issues, pros, and cons of the paper.

Best,
Your AE

---

> ### Author Response · Authors · 2022-11-15
> **Authors' general response**
>
> We thank all reviewers for their thoughtful and helpful comments. We have updated our paper accordingly. In the newer version, we have:
> - Added MoCo-V2 experiments in Table 1-4 (reviewer devn).
> - Added more explanation in hyperparameters in Appendix C.2 (reviwer 4z4q and VJn3).
> - Added more references on SSL methods trained on non-ImageNet benchmarks (reviewer 4z4q).
> - Added experiment details on computation speed and memory consumption in Appendix B (reviewer devn).
> - Added explanation how k is chosen (reviewer 4z4q).
>
> All changes are highlighted with brown color. We will address individual comments under the original reviews.

---

### Decision · Action_Editors · 2022-12-21

**Recommendation:** Reject

**Comment:**

This manuscript presents an online unsupervised learning algorithm to simultaneously learn visual representations of steaming visual data and carry out few-shot learning for newly coming categories. The core ideas are to: 1) keep and sequentially update a prototype-based memory network to allow an approximation to nonstationary environments (i.e., non-I.I.D. settings), and 2) use a contrastive loss to fulfill representation learning in context of prototypes. Two reviewers are leaned to accept the paper, whereas one reviewer has reservation.

The AE notices that Reviewer VJn3 strongly requests the experimental comparison of the presented online unsupervised learning algorithm against well-established self-supervised learning methods. The AE concurs with Reviewer VJn3 on that request, and recommends Reject since the experiments to be added cannot be done in a short period.

Nonetheless, the AE strongly suggests the authors to re-submit their significantly strengthened work to TMLR. Specifically, the AE  mandates the authors to supplement vital experiments to show: 1) the full view of comparisons between state-of-the-art self-supervised learning methods (e.g., SimCLR, SwAV, MoCo, etc.) and the algorithm proposed in this paper with varying batch sizes under the same non-I.I.D. settings, and 2) a new set of results on more large-scale and diverse image datasets.


**Audience:**

Yes.

**Claims And Evidence:**

Yes.